# Parallel Sampling from Masked Diffusion Models via Conditional Independence Testing

**Iskander Azangulov** [*]
Department of Statistics, Oxford University

**Teodora Pandeva, Niranjani Prasad, Javier Zazo, Sushrut Karmalkar**
Microsoft Research Cambridge
`{tpandeva, niranjani.prasad, javierzazo, skarmalkar}@microsoft.com`
`iskander.azangulov@spc.ox.ac.uk`

## Abstract

Masked diffusion models (MDMs) offer a compelling alternative to autoregressive models (ARMs) for discrete text generation because they enable parallel token sampling, rather than sequential, left-to-right generation. This means potentially much faster inference. However, effective parallel sampling faces two competing requirements: (i) simultaneously updated tokens must be conditionally independent, and (ii) updates should prioritise high-confidence predictions. These goals conflict because high-confidence predictions often cluster and depend on each other, opportunities for parallel updates.

We present PUNT, a model-agnostic sampler that reconciles this trade-off. Our method identifies token dependencies and removes lower-confidence tokens from conflicting groups. This produces sets of indices for unmasking that satisfy both independence and confidence criteria. Our approach ensures improved parallel unmasking through approximate conditional independence testing.

Our experiments show that PUNT delivers a superior trade-off between accuracy and compute when compared to other strong training-free baselines, especially for generation of longer sequences. On the IFEval benchmark, it achieves up to 16% higher accuracy over baseline methods, including sequential generation (one-by-one). These gains hold across different values of hyperparameters, mitigating the need for brittle hyperparameter tuning. Moreover, we observe that PUNT induces an emergent hierarchical generation strategy, where the model first establishes high-level paragraph structure before local refinement, suggesting a planning-like generation process that contributes to strong alignment performance.

## 1 Introduction

The widespread deployment of Large Language Models (LLMs) has created massive computational workloads, consuming significant datacenter resources and electricity, thereby incurring substantial operational costs. A primary driver of this inefficiency is inference speed, which is bottlenecked by the sequential, left-to-right generation process inherent in standard autoregressive models. To overcome this, alternative methods have been developed to enable multiple tokens to be generated simultaneously.

Among approaches with the potential for parallel decoding, Masked Diffusion Models (MDMs) have emerged as a particularly promising framework (Austin et al., 2023; Lou et al., 2024; Shi et al., 2024; Nie et al., 2025b). Unlike autoregressive models, MDMs iteratively refine masked sequences by predicting *subsets* of positions simultaneously, enabling parallel decoding. However, determining which tokens to unmask in parallel without degradation in quality remains challenging.

Various inference strategies have been proposed to accelerate MDMs, including confidence-based token selection (Sahoo et al., 2024; Patel et al., 2025), structured unmasking patterns (Luxembourg

---

[*]Work done during an internship at Microsoft Research Cambridge.

et al., 2025; Arriola et al., 2025), remasking (Wang et al., 2025), and distillation (Zhu et al., 2025b). However, these approaches share a critical limitation: they do not explicitly test for inter-token interference during parallel decoding. Structured patterns impose rigid, data-agnostic schedules that ignore sequence-specific dependencies, while remasking and distillation either add computational overhead or require expensive retraining.

**Our Contribution.** We propose a different approach to parallel decoding based on *contextual independence* —testing whether tokens can be decoded in parallel by checking for independence at the sampled point, rather than for all possible outcomes. Unlike standard conditional independence, which requires integrating over all possible outcomes (which is computationally prohibitive for large token spaces), contextual independence provides the part that matters at the current decoding step.

To find the contextually independent subsets, we propose PUNT (Parallel Unmasking with Non-influence Tests), a training-free procedure that employs a divide-and-conquer strategy. Our algorithm selects "anchor" subsets and tests entire "candidate" groups for dependence in batch. By carefully designing splits, PUNT certifies a large block of tokens for parallel generation using only $O(\log m)$ model calls per step (compared with $m$ for fully sequential unmasking) where $m$ is the number of masked tokens.

PUNT enjoys the following advantages: first, PUNT is **training-free** and requires no model fine-tuning or distillation. Second, unlike rigid structured patterns or confidence-based approaches, PUNT **dynamically adapts** to sequence-specific dependencies. For instance, we see that it exhibits an emergent **hierarchical generation** strategy, where the model first establishes high-level paragraph structure before refining the details. Third, for long-form text generation on alignment benchmarks as well as *de novo* protein generation tasks, PUNT outperforms other standard baselines and quickly reaches its maximum quality with very few forward passes when compared to other algorithms, resulting in a **stable Pareto frontier** over the number of forward evaluations of the MDM.

**Organization.** Section 2 introduces masked diffusion models and formalizes the parallel decoding problem. Section 3 presents our main algorithmic contribution, Section 4 presents empirical evaluation, and Section 5 discusses related work. Finally, Section 6 discusses implications and future directions.

## 2 BACKGROUND ON MASKED DIFFUSION MODELS

In this section, we review the fundamentals of masked diffusion models and establish the notation used throughout this paper.

**Notation.** We denote vectors with bold lowercase (e.g., $\mathbf{x}$) and scalars with regular lowercase (e.g., $y$); random variables use uppercase (e.g., $X, \mathbf{Y}$) with corresponding lowercase for their realizations (e.g., $x, \mathbf{y}$). We will also use uppercase letters to denote sets and tensors, when it is clear from context. Let $L = [\ell] := \{1, \ldots, \ell\}$ denote integers up to $\ell$. For $I \subseteq L$, $-I := L \setminus I$ is its complement, and $\mathbf{x}^I = \{x^i \mid i \in I\}$ represents the indexed subset of sequence $\mathbf{x} = (x^1, \ldots, x^\ell)$. With vocabulary $V$, we consider discrete state space $V^L$. For random sequence $\mathbf{X} = (X^1, \ldots, X^\ell)$, we write $p(\mathbf{x}) := \boldsymbol{P}(\mathbf{X} = \mathbf{x})$ for outcome $\mathbf{x} \in V^L$, and $p^j(\cdot)$ for the marginal at position $j$. Extending to $V_{\text{MASK}} = V \cup \{\text{MASK}\}$, a token $x^i$ is **masked** if $x^i = \text{MASK}$. For any sequence, $M = M(\mathbf{x}) := \{i \mid x^i = \text{MASK}\}$ denotes masked indices, with unmasked indices denoted $-M$. Conditional distribution at position $j$ given observed tokens $\mathbf{x}^I$ is written as $p^j(\cdot \mid \mathbf{x}^I)$, shorthand for $\boldsymbol{P}(X^j = \cdot \mid \mathbf{X}^I = \mathbf{x}^I)$. For a sequence with masked indices $M$, $\mathbf{x}^{-M}$ denotes unmasked tokens. In the iterative generation process, $\mathbf{x}_t$ represents the sequence at step $t$. and $M_t := M(\mathbf{x}_t)$ denotes masked indices at step $t$.

**Masked Language Modeling.** Masked language modeling trains neural networks to predict missing tokens from context. Given a sequence $\mathbf{x} \in V_{\text{MASK}}^\ell$ with masked positions $M$, the model parameterized by $\theta$ learns conditional distributions:

$$p_\theta^i(\cdot \mid \mathbf{x}^{-M}) \approx p^i(\cdot \mid \mathbf{x}^{-M}), \quad \forall i \in M.$$

During training, a clean sample (i.e. without any masked coordinates) $\mathbf{x}_{\text{clean}} \sim p(\cdot)$ is drawn from the true data distribution. A random subset of its tokens $M \subseteq L$ is then selected to be masked, creating the corrupted sequence $\mathbf{x}$ where $\mathbf{x}^{-M} = \mathbf{x}_{\text{clean}}^{-M}$ and $\mathbf{x}^M$ consists of MASK tokens.

The model parameters $\theta$ are optimized to maximize the conditional log-likelihood of the original tokens at masked positions:

$$\mathcal{L}(\theta) = \mathbb{E}_{\mathbf{x}_{\text{clean}}, M} \left[ \sum_{i \in M} \log p_\theta^i(x_{\text{clean}}^i \mid \mathbf{x}^{-M}) \right].$$

Generation proceeds iteratively using the trained model. Starting from a fully masked sequence $\mathbf{x}_0 = (\text{MASK}, \ldots, \text{MASK})$, each iteration performs two operations at timestep $t$: (1) sample candidate tokens $y_t^i \sim p_\theta^i(\cdot \mid \mathbf{x}_t^{-M_t})$ for all masked positions $i \in M_t$, and (2) update a subset $R \subseteq M_t$ of these positions with their sampled values, so that $x_{t+1}^i \leftarrow y_t^i$ for $i \in R$ and $x_{t+1}^i \leftarrow x_t^i$ for $i \notin R$. This process repeats until all positions are unmasked, producing the final sequence $\mathbf{x}_T$.

The iterative sampling process introduces two key sources of error at each step that can compromise sample quality. For clarity, we analyze the error within a single step and drop the time index $t$ in the following discussion.

**Approximation Error.** The learned model $p_\theta^i(\cdot \mid \mathbf{x}^{-M})$ only *approximates* the true conditional distribution $p^i(\cdot \mid \mathbf{x}^{-M})$. This potentially leads to suboptimal token predictions. To mitigate this, we employ a confidence score $\phi_i$ per position $i$, to guide mask selection at each step, updating only those positions where the model exhibits high confidence. This strategy improves generation quality by prioritizing high-confidence predictions. Common confidence scores $\phi_i$ for position $i$ include: **negative entropy** $\sum_{x \in V} p_\theta^i(x \mid \mathbf{x}^{-M}) \log p_\theta^i(x \mid \mathbf{x}^{-M})$, **confidence** $p_\theta^i(y^i \mid \mathbf{x}^{-M})$ of the sampled token, and **top margin** between the two most likely tokens.

**Joint Dependencies.** A more fundamental limitation arises from the sampling strategy itself. When sampling masked tokens $y^i$ independently from their conditional distributions $p_\theta^i(\cdot \mid \mathbf{x}^{-M})$, we implicitly assume conditional independence among all masked tokens given the unmasked context. Natural sequences, however, exhibit complex dependencies that violate this assumption. True conditional independence for candidate tokens $R \subseteq M$ requires the joint probability to factorize as:

$$p^R(\cdot \mid \mathbf{x}^{-M}) = \prod_{i \in R} p^i(\cdot \mid \mathbf{x}^{-M}) \tag{1}$$

In general, the joint distribution does not factorize in this way. Finding a subset of tokens where this factorization holds presents significant computational challenges, as verifying this condition requires checking that equation 1 holds for all outcomes $\mathbf{y}^R \in V^R$—a space that grows exponentially with $|R|$. The following section presents an efficient method to identify token subsets that approximately satisfy this independence condition, thereby mitigating this source of error.

## 3 METHOD

In this section, we introduce our method for one step of parallel unmasking. We first establish **contextual independence** as the criterion for safe parallel unmasking (§3.1), then present our **efficient subset discovery** algorithm that identifies independent token sets using only $O(\log |M|)$ model evaluations (§3.2).

### 3.1 CONTEXTUAL INDEPENDENCE

To address joint dependencies, we adopt the notion of *contextual independence* as our criterion for parallel unmasking. This property precisely characterizes when parallel sampling yields the same distribution as sequential sampling. Unlike full statistical independence (overly restrictive) or confidence-based heuristics (which ignore dependencies), contextual independence identifies tokens that can be unmasked simultaneously given the current context.

**Definition 3.1** (Contextually Independent Random Variables). A random variable $X$ is *contextually independent* of a random variable $Y$ at a point $y$ if the conditional distribution of $X$ given $Y = y$ is identical to the marginal distribution of $X$, i.e., $p_{X|Y}(\cdot \mid Y = y) = p_X(\cdot)$.

**Definition 3.2** (Contextually Independent Sequences). A *sequence* of random variables $(X^1, \ldots, X^\ell)$ is *contextually independent* at an outcome $(x^1, \ldots, x^\ell)$, if for each $i \in L$, the conditional distribution of $X^i$ given the preceding outcomes $\mathbf{x}_{<i} = (x^1, \ldots, x^{i-1})$ is identical to its marginal distribution. Formally, for all $i \in L$: $p_{X^i|\mathbf{X}_{<i}}(\cdot \mid \mathbf{x}_{<i}) = p_{X^i}(\cdot)$, where $\mathbf{X}_{<i} = (X^1, \ldots, X^{i-1})$.

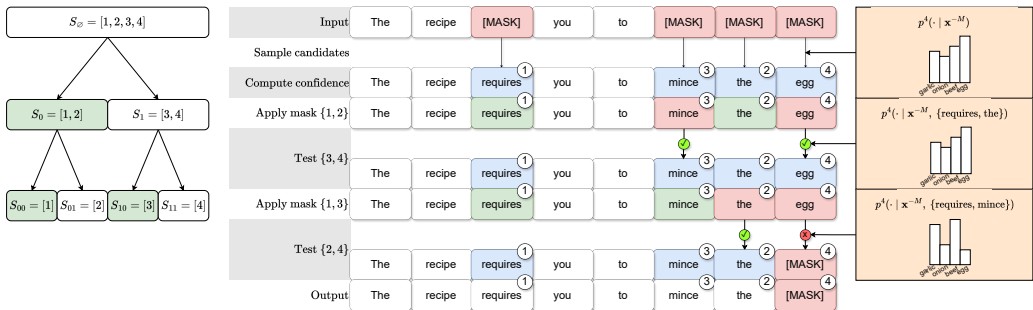

Figure 1: Illustration of one iteration of PUNT on 4 masked tokens, which consists of $\log_2 4 = 2$ tests. Left: A binary tree representing the recursive partitioning. Each level corresponds to a single parallel test in the iterative algorithm. Right: In each round, confidence-ordered tokens (circled numbers) are partitioned into "anchor" (green) and "test" sets. Test tokens that are dependent on the anchor set are rejected (red), while independent ones (blue) are kept. Each token must pass all independence tests to be accepted. Here, "mince" passes (independent of {"requires", "the"}) while "egg" fails (dependent on {"requires", "mince"}). The final set {"requires", "the", "mince"} satisfies contextual independence: $p(\text{"requires"}, \text{"the"}, \text{"mince"} \mid \mathbf{x}_{\text{unmasked}}) = p(\text{"requires"} \mid \mathbf{x}_{\text{unmasked}}) \cdot p(\text{"the"} \mid \mathbf{x}_{\text{unmasked}}) \cdot p(\text{"mince"} \mid \mathbf{x}_{\text{unmasked}})$.

In other words, under the contextual independence assumption, sampling the vector $(x^1, \ldots, x^\ell)$ sequentially is equivalent to sampling its components in parallel.

Our goal is, given a candidate vector $\mathbf{y}^M$, to find an *ordered* [1] set of masked indices $R = \{r_1, \ldots, r_{|R|}\} \subseteq M$ that are contextually independent relative to the unmasked context $\mathbf{x}^{-M}$. Formally, for any $i \in \{1, \ldots, |R|\}$, the distribution at position $r_i$ must be independent of the outcomes at preceding positions $R_{<i} := \{r_j \mid j < i\}$, given the unmasked context:

$$p^{r_i}(\cdot \mid \mathbf{x}^{-M}, \mathbf{y}^{R_{<i}}) = p^{r_i}(\cdot \mid \mathbf{x}^{-M}). \tag{2}$$

A naive, greedy approach to construct such a set would be to iterate through all masked indices $m \in M$ and sequentially add an index to $R$ if it satisfies Equation 2 given the previously added indices. However, this requires $O(|M|)$ sequential model evaluations, which defeats the purpose of parallel sampling.

We propose an efficient recursive algorithm based on a recursive divide-and-conquer strategy, which we will later provide an an efficient iterative implementation for. The validity of this approach relies on the following stability assumption regarding the conditional independence structure of the model.

**Assumption 3.3.** (Independence Stability) Let $i \in M$ be a masked index, and let $U \subseteq M \setminus \{i\}$ be a subset of masked indices. If for some sequence of tokens $\mathbf{y}^U$ we have $p^i(\cdot \mid \mathbf{y}^U, \mathbf{x}^{-M}) = p^i(\cdot \mid \mathbf{x}^{-M})$, then for any $W \subset U$ it holds that $p^i(\cdot \mid \mathbf{y}^W, \mathbf{x}^{-M}) = p^i(\cdot \mid \mathbf{x}^{-M})$.

This assumption represents a balanced compromise between complete independence and simple contextual independence. It states that if a set of positions $U$ does not influence the prediction at position $i$, then any subset $W \subset U$ will also not influence that prediction. This property ensures that independence tests conducted at any stage of our recursive algorithm remain valid throughout all subsequent stages. Section B provides a justification for why this assumption is reasonable for transformer-based architectures and empirical evidence that it approximately holds in practice.

### 3.2 EFFICIENT SUBSET DISCOVERY

Under Assumption 3.3, we can construct the set $R$ in $O(\log |M|)$ parallel steps. If there is at least one masked position (i.e. $|M| \geq 1$), the recursive algorithm starts with $S = M$ and proceeds as follows (see Figure 1 for an illustration of one iteration):

---

[1]Eventually we will be ordering these via confidence metrics $\phi_i$, see Section 3.3.

At each recursive call, its input is a (confidence) [2] ordered subset of masked candidates $S = (s_1, s_2, \ldots, s_{|S|}) \subseteq M$. The base case for the recursion is when $|S| \leq 1$, in which case the procedure returns $S$. For larger sets, the algorithm proceeds as follows:

1. **Divide:** The ordered input set $S$ is split into two balanced halves: the "anchor" set $S_0 = (s_1, \ldots, s_p)$ and the "test" set $S_1 = (s_{p+1}, \ldots, s_{|S|})$, where $p$ is a split point of the designer's choice.

2. **Prune** (Filter)**:** The "test" set $S_1$ is pruned based on its dependency on the candidates $\mathbf{y}^{S_0}$. For each index $i \in S_1$, we compute its new conditional distribution and measure the change from the baseline using the KL divergence:
$$\varepsilon_i := D_{\mathrm{KL}}\big(p^i(\cdot \mid \mathbf{x}^{-M}) \,\big\|\, p^i(\cdot \mid \mathbf{x}^{-M}, \mathbf{y}^{S_0})\big)$$
A filtered set $S_1'$ is then formed by retaining only those indices for which the change is below a threshold $\varepsilon > 0$: $S_1' = \{i \in S_1 \mid \varepsilon_i < \varepsilon\}$.

3. **Recurse:** The algorithm then makes two independent (parallel) recursive calls: one on the "anchor" set $S_0$ and another on the filtered "test" set $S_1'$ and obtains $R_0$ and $R_1$ respectively.

4. **Combine:** The final result $R$ for the input set $S$ is the **union** (ordered sum) of the outputs $R := R_0 \sqcup R_1$ from the two recursive calls above. Note that by construction any token in $R_1$ is contextually independent of $S_0$ and by Assumption 3.3, it is contextually independent of subset $R_0 \subset S_0$.

Choosing $p = \lfloor |S|/2 \rfloor$ for each recursive iteration, we ensure that the recursion depth is $O(\log |M|)$. In the next section, we discuss how to execute all calls at the same recursion level using a single network evaluation, thereby achieving $O(\log |M|)$ cost per round.

### 3.3 CONFIDENCE ALIGNMENT AND IMPLEMENTATION DETAILS

This section addresses two key implementation aspects: (i) incorporating confidence-based prioritization into our recursive algorithm to maintain generation quality, and (ii) transforming the recursive procedure into an efficient iterative implementation.

**Confidence-Ordered Splits.** At each recursive step, the candidate set $S$ is split into $S_0$ and $S_1$, and positions in $S_1$ are pruned if they exhibit strong dependence on $S_0$. By sorting the initial candidate set $S$ in descending order of confidence (see Section 2 for options), $\phi_{s_1} > \phi_{s_2} > \cdots > \phi_{s_{|S|}}$, we ensure that $S_0$ always contains tokens with at least median-level confidence. Consequently, tokens pruned from $S_1$ necessarily have lower confidence than those retained in $S_0$. In fact, this also ensures that during each unmasking step, the highest-confidence token in $M$ is always included in the final set $R$, since it will never be pruned.

**Binary Encoding of Recursive Calls.** To enable parallel computation, we transform our recursive algorithm into an efficient iterative procedure. At each level of the recursion tree, we combine all independence tests into a single, parallel model evaluation.

To see how this might be done, suppose at some recursion level we test pairs $(S_0^1, S_1^1), (S_0^2, S_1^2), \ldots (S_0^k, S_1^k)$, by construction, these sets form a partition of a subset of $M$. We propose, instead of performing these tests independently, to test all "test" tokens against the union of all "anchor" sets $\bigsqcup_\ell S_0^\ell$. Then, Assumption 3.3 ensures that passing this combined test implies passing the individual tests. Formally, for any $\ell$ and any $i \in S_1^\ell$ if $p_\theta^i(\cdot \mid \mathbf{y}^{\sqcup S_0^\ell}, \mathbf{x}^{-M}) = p_\theta^i(\cdot \mid \mathbf{x}^{-M})$, then it also satisfies $p_\theta^i(\cdot \mid \mathbf{y}^{S_0^\ell}, \mathbf{x}^{-M}) = p_\theta^i(\cdot \mid \mathbf{x}^{-M})$.

*Binary Representation of Recursive Splits.*

We now describe how this idea can be employed to convert our recursive algorithm to an iterative one. The recursive splits are determined by each token's position in the confidence-ordered list $M$. This allows us to pre-determine all splits by assigning a binary code to each position. More precisely, we assign each position $i \in \{1, \ldots, |M|\}$ a binary representation $\mathrm{bin}(i)$ with $\lceil \log_2 |M| \rceil$ bits, padded with zeros if necessary. Tracking this binary encoding allows us to identify the path of each node in the recursion tree.

---

[2]See Section 3.3 for details

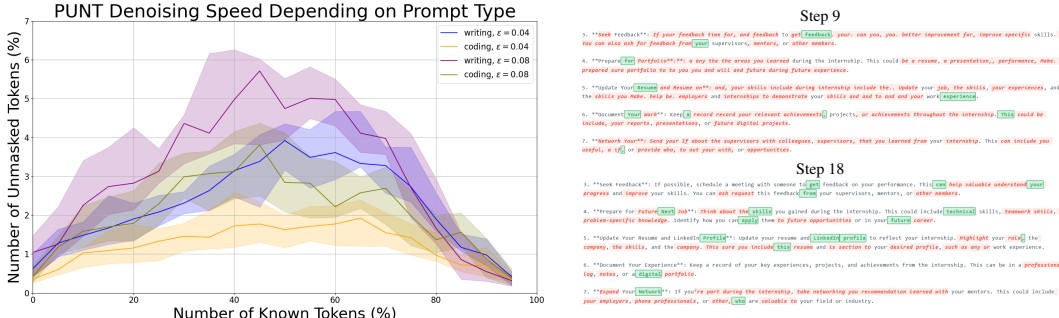

Figure 2: Left: Unmasking efficiency for various prompt types vs number of unmasked tokens. Right: Visualization of the denoising process at steps 9 and 18 for a sample prompt ("What should I do at the end of the internship."). Tokens are color-coded: green tokens are accepted by PUNT for parallel unmasking in the current step, red tokens are rejected, and uncolored tokens were unmasked in previous steps. See Appendix D for more examples of intermediate denoising steps.

At recursion level $b$, we have $2^b$ nodes, with each node indexed by a unique binary prefix $\mathbf{u} \in \{0,1\}^b$ corresponding to the subset $S_{\mathbf{u}} = \{i \le |M| : \text{prefix}_b[\text{bin}(i)] = \mathbf{u}\}$, which is then partitioned into "anchor" ($S_{\mathbf{u}0}$) and "test" ($S_{\mathbf{u}1}$) subsets (see Figure 1 (Left) for an example).

We would like to combine all $2^b$ tests at recursion level $b$ into a single test. To do so, we define a global partition for level $b$ based on the $b$-th bit of the binary encoding. $B_b = \{i \in [|M|] : \text{the } b\text{-th bit of bin}(i) = 0\}$.

Starting with $R = M$, each round $b$ can now partition the current set using the predefined binary split $B_b$: the "anchor" tokens ($S_0 = R \cap B_b$) and "test" tokens ($S_1 = R \setminus B_b$). All tokens in $S_1$ are tested for dependence on $\mathbf{y}^{S_0}$ in *a single forward pass*, dependent tokens are removed from $R$. After all $\log|M|$ rounds complete, the remaining set $R$ contains only contextually independent tokens.

The resulting procedure, summarized below (and in Algorithm 1), requires only $O(\log|M|)$ forward evaluations of the model per denoising step and guarantees that the returned set $R$ consists of contextually independent, high-confidence tokens that can be unmasked in parallel.

*Iterative Algorithm.* Given confidence-ordered masked tokens $M = \{m_1, m_2, \ldots, m_{|M|}\}$ where $\phi_{m_1} \ge \phi_{m_2} \ge \cdots \ge \phi_{m_{|M|}}$, we initialize $R \leftarrow M$ and execute $\lceil \log_2 |M| \rceil$ iterations.

For each iteration $b \in \{1, \ldots, \lceil \log_2 |M| \rceil\}$:

1. **Test:** Partition $R$ into anchor tokens $S_0 = R \cap B_b$ and test tokens $S_1 = R \setminus B_b$.

2. **Prune:** For each $j \in S_1$, compute the KL divergence $d_j = D_{\text{KL}}(p^j(\cdot \mid \mathbf{x}^{-M}) \,\|\, p^j(\cdot \mid \mathbf{x}^{-M}, \mathbf{y}^{S_0}))$ in a single forward pass.

3. **Update:** Remove dependent tokens from $R$: $R \leftarrow R \setminus \{j \in S_1 : d_j > \varepsilon\}$.

The final set $R$ contains tokens that can be unmasked in parallel without interfering with each other. *Remark* 3.4 (Conservative but Parallel Testing). The iterative implementation uses *batched* independence tests: tokens at each tree level are tested against the union of all anchor tokens at that level, not just their specific recursive subset. This stricter condition enables full parallelization. Empirically, this conservative approach maintains strong performance on long-context tasks while significantly reducing runtime.

### 3.4 ALGORITHMIC PROPERTIES AND INDEPENDENCE STABILITY

This section discusses PUNT's properties for text generation and justifies Assumption 3.3.

**Adaptive Unmasking.** Our sampler exhibits emergent hierarchical generation, first establishing high-level structure (e.g., paragraphs, headings) before filling in details, as shown in Fig. 2 (right). As can be seen, at step 9, the model has already generated the main headings and subheadings of

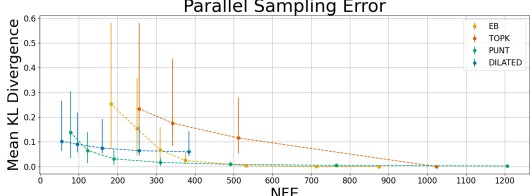 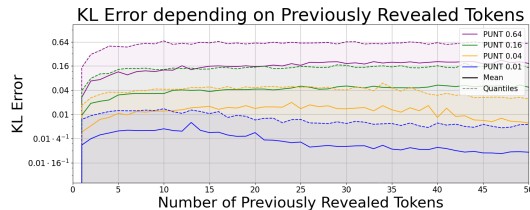

Figure 3: Parallel sampling error (equation 3). Left: Average error for different samplers compared to number of forward evaluations (NFEs). Right: Median together with confidence intervals $(Q_5, Q_{95})$ of the error for PUNT samplers with different $\varepsilon$ as a function of the number of previously revealed tokens. Note that $Q_5$ remains below $10^{-3}$ across all positions.

the article, while the rest of the text remains masked. By step 18, the model has begun filling in the details under each heading.

We hypothesize this behavior stems from the conditional independence between high-level structural tokens and fine-grained details. Because the latter exert minimal influence on the former, the structural tokens pass the independence tests in PUNT's filtration stage and are unmasked early. A formal investigation of this phenomenon is left for future work.

This hierarchical generation has a cascading effect. Once revealed, high-level tokens act as contextual anchors, partitioning the text into conditionally independent sections. This allows the sampler to unmask tokens in different sections in parallel, adapting its denoising speed to the task's inherent structure. As shown in Fig. 2 (left), this results in different performance profiles for various prompts.

**Independence Stability of Transformers.** Assumption 3.3 (Independence Stability) is a direct consequence of the Transformer architecture's attention mechanism. In Transformers, the influence of one token on another is governed by attention weights; if the attention from position $i$ to position $j$ is zero, then position $j$ has no direct influence on the representation at position $i$.

This relationship between attention and influence allows us to connect conditional independence to attention scores. Specifically, we argue that a token $y_i$ is conditionally independent of a set of tokens $\mathbf{y}^V$ given the remaining tokens $\mathbf{x}^{-M}$ if and only if the total attention from position $i$ to all positions in $R$ is negligible across all layers and heads, i.e.,

*The conditional distribution $p_\theta^i(\cdot \mid \mathbf{y}^R, \mathbf{x}^{-M})$ equals $p_\theta^i(\cdot \mid \mathbf{x}^{-M})$ if and only if[3] the cumulative attention weight from position $i$ to all positions in $R$ is negligible.*

This property directly implies Independence Stability. Since attention weights are non-negative, if the attention from position $i$ to a set $R$ is negligible, the attention to any subset $U \subset R$ must also be negligible. A detailed justification is provided in Appendix B.

## 4 EXPERIMENTS

We evaluate our proposed sampler, PUNT, on a number of natural language tasks. Our empirical results validate the effectiveness of our approach and support the attention hypothesis introduced in Appendix B. We evaluate: (i) PUNT's performance on long-form text generation tasks such as MTBench; (ii) PUNT's effectiveness on short-answer benchmarks for mathematics and code generation; (iii) The error introduced by parallel token sampling and its relationship to the exploration rate $\varepsilon$; (iv) Deviation in empirical attention patterns (supporting theoretical independence assumptions).

**Experimental Setup.** We evaluate PUNT on two powerful, open-source large language models: Dream 7B (Ye et al., 2025a) and LLaDA 1.5 (Zhu et al., 2025a).

**Baselines.** We compare PUNT against three strong, training-free baseline samplers. These baselines include: (i) standard top-$k$ sampling; (ii) the EB-sampler (Patel et al., 2025); and (iii) the

---

[3]Except for padding end-of-sequence(EOS) tokens, see details in Appendix B

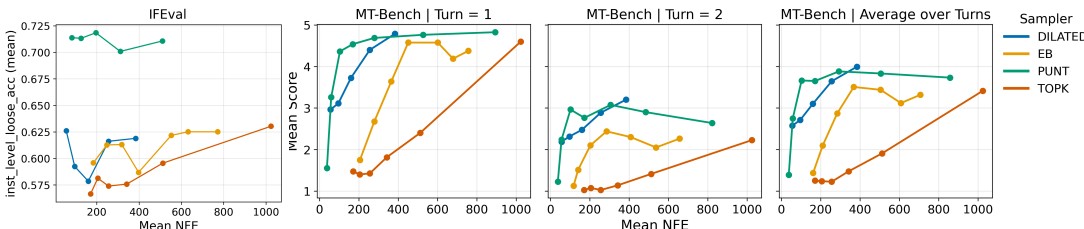

Figure 5: IFEval and MTBench performance of PUNT compared to baselines on Dream 7B. Benchmark specific scores (higher is better) vs mean number of forward passes.

Dilated-sampler (Luxembourg et al., 2025). Note that all the samplers are implemented for a given context length $|M|$, in a **non-semi-autoregressive** manner and the exact parameter configurations are provided in Appendix C.3.

We also evaluate the performance of PUNT on generation in a structured biological domain. Specifically, we look at unconditional generation of *de novo* membrane proteins using MemDLM (Goel et al., 2024), a state-of-the-art protein language model. Full details are provided in Appendix C.5.

We provide additional analysis in the appendices, including performance-vs-$\epsilon$ plots (Appendix E), experiments with varying $\epsilon$ schedules across generation trajectories (Appendix F), and comparisons and discussion with APD (Appendix C.4), proposed by Israel et al. (2025).

## 4.1 ALIGNMENT BENCHMARKS

We evaluate PUNT on the instruction-following benchmarks MTBench (Bai et al., 2024) and IFEval (Zhou et al., 2023). MTBench comprises 80 tasks across diverse domains—including creative writing, logical reasoning, and code generation—providing a robust evaluation framework for model performance. Each task consists of two turns, with the second turn depending on the output of the first. IFEval complements this by specifically measuring instruction-following accuracy through a collection of carefully designed test cases that evaluate precise adherence to complex instructions. Full benchmark details are provided in Appendix C.1. As shown in Figure 5, PUNT consistently outperforms all baseline samplers on both benchmarks, achieving higher scores across both metrics (`inst_level_loose_acc` and mean score; higher is better). Importantly, it delivers these gains while requiring substantially fewer forward evaluations (NFEs).

**Note on NFE regimes:** We see that PUNT's primary advantage lies in the low-to-mid NFE regime on long-form generation tasks. Since each PUNT iteration requires $\lceil \log_2 |M| \rceil$ forward passes for independence testing (e.g., 10 passes for 1024 tokens), the method excels when the NFE budget allows for meaningful independence checking while still providing efficiency gains. At very high NFE budgets (e.g., NFE $\geq 400$ on MT-Bench), curves may converge or cross as fixed-geometry schedulers like Dilated can afford many denoising steps, while PUNT's overhead from independence testing becomes proportionally larger.

Figure 4: MBPP performance of PUNT compared to baselines on Llada. MBPP pass@1 (higher is better) vs mean number of forward passes.

## 4.2 SHORT-ANSWER BENCHMARKS

We evaluate PUNT across diverse benchmarks spanning mathematical reasoning (GSM8K (Cobbe et al., 2021)) and code generation (HumanEval (Chen et al., 2021), MBPP (Austin et al., 2021)).

As expected, PUNT underperforms on short-answer tasks with limited context since it requires multiple forward passes for complete generation. For instance, in the MBPP benchmark on LLaDA with temperature 0.7 (see Fig. 4), PUNT's performance aligns closely with EB when evaluated

by the number of forward evaluations (NFE). However, when measured by the number of denoising steps—that is, the number of algorithmic steps, where each step may involve multiple forward passes in parallel—PUNT outperforms other samplers. A simple fix could be to use PUNT only for the latter part of the generation, but we leave this for future work. Similar trends are observed across other benchmarks in this group. Results comparing all samplers, models, and hyperparameters appear in Appendix C.3.

## 4.3 PUNT Sampler Error Analysis

We empirically quantify the parallel sampling error on the LLaDA model. For this, we generate 1024-token responses for MTBench (Bai et al., 2024) with exploration rates $\varepsilon \in \{0.01, \ldots, 0.32\}$. Within each parallel generation step, tokens are ordered by confidence before sampling. For the $i$-th token at position $r_i$ unmasked in parallel, we compute the error between the true conditional distribution and our independence approximation:

$$\delta_{\mathrm{KL}}^{r_i} = D_{\mathrm{KL}}\big(p_\theta^{r_i}(\cdot \mid \mathbf{x}^{-M}) \, \big\| \, p_\theta^{r_i}(\cdot \mid \mathbf{x}^{-M}, \mathbf{y}^{R<i})\big). \tag{3}$$

This KL divergence quantifies the information lost by assuming token $r_i$ is conditionally independent of other tokens unmasked in the same step, $\mathbf{y}^{R<i}$. As shown in Fig. 3, PUNT achieves a low parallel sampling error while maintaining a small NFE compared to other samplers. Furthermore, the parallel decoding error, $\delta_{\mathrm{KL}}$, remains robustly below the $\varepsilon$ threshold, irrespective of the number of tokens previously revealed in the step.

## 5 Related Work

Our work builds on recent advances in discrete diffusion models and inference-time planners.

**Masked Diffusion Models (MDMs).** Discrete diffusion models (Austin et al., 2023) offer a non-autoregressive alternative for text generation. Training objectives based on score matching (Lou et al., 2024) and masked language modeling (Sahoo et al., 2024) have enabled large-scale models like LLaDA (Nie et al., 2025b) and others (Nie et al., 2025a). Commercial implementations include Gemini Diffusion (Google DeepMind, 2025) and Mercury (Inception Labs, 2025). MDMs face two key limitations: compounding errors from parallel unmasking and inefficient KV caching. We address the former by identifying token sets for safe parallel unmasking, which minimizes interference and improves both efficiency and quality.

**Inference-Time Planners for Acceleration.** Efficient inference scheduling remains MDMs' central challenge. Various training-free planners aim to minimize function evaluations (NFEs) while maintaining generation quality.

*Confidence and Entropy Gating.* Confidence-based scheduling iteratively unmasks tokens with highest model confidence (or lowest entropy) (Sahoo et al., 2024). The EB-Sampler extends this by dynamically unmasking variable-sized token sets whose aggregate entropy stays below threshold $\gamma$ (Patel et al., 2025). While adaptive, these methods remain conservative, ignore token independence, and typically unmask only small subsets.

*Remasking and Refinement.* Several methods correct parallel decoding errors through remasking. ReMDM (Wang et al., 2025) iteratively remasks and updates generated tokens. Path-Planning (P2) (Peng et al., 2025) and DDPD (Liu et al., 2025) separate inference into planning (selecting tokens to update/remask) and denoising stages. While improving quality, these approaches increase NFE through corrective passes.

*Spacing Schedulers.* These fixed-geometry (non-adaptive) methods enforce spatial separation between parallel unmaskings. Dilated scheduling unmasks non-adjacent token groups for improved stability (Luxembourg et al., 2025). Halton-based schedulers use low-discrepancy sequences for uniform spacing (Besnier et al., 2025). Block Diffusion balances AR and parallel generation by processing contiguous spans (Arriola et al., 2025).

*Analysis of Ordering and Scheduling.* Recent theoretical and empirical work has deepened the community's understanding of these schedulers. Kim et al. (2025) study the impact of token ordering, showing that adaptive inference can sidestep computationally hard subproblems. Park et al. (2024) focus on optimizing the temporal schedule (the number and placement of diffusion steps) to reduce

NFEs. Others have explored MDLMs for complex reasoning, where planning is critical (Ye et al., 2025b), and for specialized domains like code generation (Gong et al., 2025).

**Comparison to Autoregressive Accelerators.** While autoregressive models like LLaMA-3 (Grattafiori et al., 2024) are accelerated by speculative decoding (Leviathan et al., 2023; Xia et al., 2022), this approach remains fundamentally sequential. In contrast, our method reduces NFEs by leveraging the non-sequential, any-order generation capabilities of MDMs. Orthogonal optimizations like KV caching are applicable to both paradigms (Ma et al., 2025; Hu et al., 2025).

## 6 CONCLUSION AND FUTURE WORK

We introduced PUNT, a training-free sampler that looks to resolve the conflict between speed and quality in MDMs by efficiently identifying sets of approximately conditionally independent tokens for parallel unmasking. This enables a significant reduction in the number of model evaluations needed for generation while preserving output quality. We provided a conceptual justification for its applicability to transformer architectures and validated its effectiveness on mathematics, code, and long-form text benchmarks. We also observe that PUNT induces an emergent hierarchical generation strategy: coarse paragraph structure is established early, followed by localized refinement.

Future work can extend this approach in several directions: (i) developing adaptive or curriculum-style schedules for the independence threshold $\epsilon$ to balance early exploration with late precision; (ii) distilling PUNT into a student model that predicts contextually independent reveal sets in a single forward pass; and (iii) combining PUNT with orthogonal efficiency techniques such as KV-caching, to further shift the accuracy–compute Pareto frontier.

**Reproducibility.** We provide detailed descriptions of our algorithms and experimental methodology throughout the paper. The core PUNT algorithm is presented in Section 3 and in Algorithm 1, with the recursive structure detailed in Section 3.2. Our implementation of binary encoding for efficient parallel pruning is described in Section 3.3.

Our experimental setup is described in Section 4, with comprehensive benchmark details provided in Appendix C.

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

# Appendix

## USE OF LARGE LANGUAGE MODELS

During the preparation of this work, the authors used large language models (LLMs), including Google's Gemini and OpenAI's GPT-5, as writing, search, and coding assistants. The use of these tools was restricted to the following tasks: improving the grammar, clarity, and readability of the text; assisting with the discovery of related literature to support our literature review process; and generating code for implementation and visualizations of experiments. The authors have reviewed and edited all model-generated content and take full responsibility for the final content of this paper, including any errors or omissions.

## A  ORGANISATION OF APPENDIX

The rest of the appendix is organized as follows. In Appendix B, we justify Assumption 3.3 by demonstrating that it holds for Transformer-based masked language models, which is a direct consequence of the Transformer's attention mechanism. In Appendix C, we provide additional experimental details and results. In Appendix C.5, we provide some preliminary experiments on a protein masked diffusion model. In Appendix D, we provide two examples of text that is generated by PUNT.

*Remark on Notation:* In addition to standard notation as defined in the paper, in the appendix, we will also use upper-case bold letters (such as $\mathbf{A}$) to denote tensors. We will use lowercase and unbolded letters to denote scalars (such as $A_{ij}$). In addition, we may have uppercase letters (such as $Q, K, V$) annotations to help annotate different matrices. This is to accommodate standard notation used in the literature.

## B  INDEPENDENCE STABILITY

In this section, we demonstrate that Assumption 3.3 holds for Transformer-based masked language models, which is a direct consequence of the Transformer's attention mechanism. Let us start with recalling the assumption.

**Assumption B.1.** (Independence Stability) Let $i \in M$ be a masked index, and let $U \subseteq M \setminus \{i\}$ be a subset of masked indices. If for some sequence of tokens $\mathbf{y}^U$ we have $p^i(\cdot \mid \mathbf{y}^U, \mathbf{x}^{-M}) = p^i(\cdot \mid \mathbf{x}^{-M})$, then for any $W \subset U$ it holds that $p^i(\cdot \mid \mathbf{y}^W, \mathbf{x}^{-M}) = p^i(\cdot \mid \mathbf{x}^{-M})$.

Next, we recall the design of attention mechanism and discuss prior works

**Attention-Based Independence.**  In transformers Vaswani et al. (2023), the attention weights control information flow between positions. For an input sequence $\mathbf{X} = (\mathbf{X}^1, \ldots, \mathbf{X}^L) \in \mathbb{R}^{L \times d_{in}}$, each attention head computes query, key, and value vectors for every position:

$$\mathbf{Q}^i = \mathbf{W}^Q \mathbf{X}^i, \quad \mathbf{K}^i = \mathbf{W}^K \mathbf{X}^i, \quad \mathbf{V}^i = \mathbf{W}^V \mathbf{X}^i,$$

where $\mathbf{W}^Q, \mathbf{W}^K \in \mathbb{R}^{d_k \times d_{in}}$ and $\mathbf{W}^V \in \mathbb{R}^{d_v \times d_{in}}$ are learned weight matrices.

The attention mechanism then computes pairwise attention scores between all positions through scaled dot products:

$$\mathbf{A} = \text{softmax}\left(\frac{\mathbf{Q}\mathbf{K}^\top}{\sqrt{d_k}}\right) \in \mathbb{R}^{L \times L},$$

where $\mathbf{Q}, \mathbf{K} \in \mathbb{R}^{L \times d_k}$ stack the query and key vectors across all positions. The attention weights $A^{ij}$ quantify how much position $j$ influences position $i$, computed via normalized dot-product similarity. The output of one head combines value vectors weighted by these attention scores: $\mathbf{head}_h = \mathbf{A}\mathbf{V} \in \mathbb{R}^{L \times d_v}$. Finally, outputs of different heads are stacked to get, $\mathbf{Z_i} = \text{concat}(\mathbf{head}_1^i, \ldots, \mathbf{head}_H^i)$. The output of the layer $\mathbf{Y} = (\mathbf{Y}^1, \ldots, \mathbf{Y}^L) \in \mathbb{R}^{L \times d_{out}}$ is calculated as $\mathbf{Y}^i = F(\mathbf{Z}^i)$ by application to each of the coordinates of MLP together with normalization layers and skip connections.

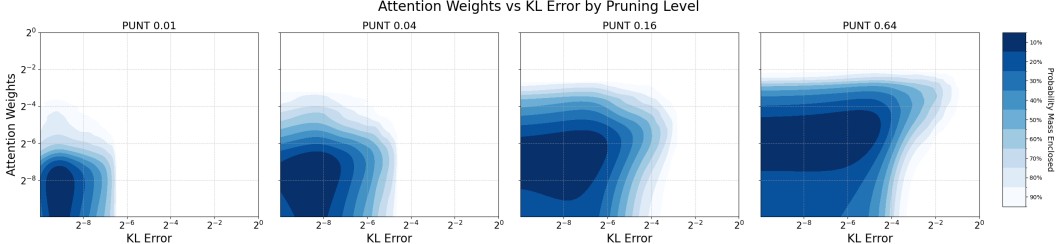

Figure 6: Joint distribution of $\delta_{\mathrm{KL}}$ – the sampling error and $\delta_A$ – the total attention to the previous tokens revealed in parallel.

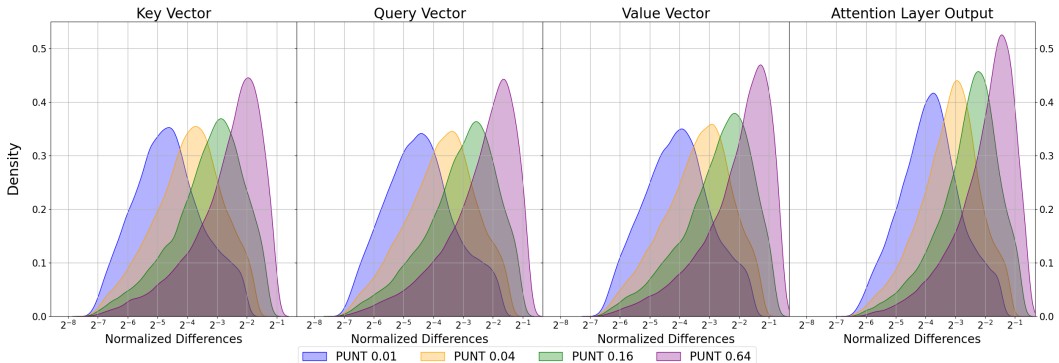

Figure 7: Difference between attention statistics for $p^{r_i}(\cdot \mid \mathbf{x}^{-M}, \mathbf{y}^{R<i})$ relative to the same statistics computed while evaluating $p^{r_i}(\cdot \mid \mathbf{x}^{-M})$.

Crucially, the attention weights $A_{ij}$ directly control information flow: when $A_{ij} = 0$, position $j$'s value vector $\mathbf{V}^j$ contributes nothing to position $i$'s output. The model's final predictions are obtained by applying softmax to the last layer's output: $p_\theta^i(\cdot \mid \mathbf{x}^{-M}) := \mathrm{softmax}(\mathbf{Y}^i)$. Therefore, $p_\theta^i(\cdot \mid \mathbf{x}^{-M}) = p_\theta^i(\cdot \mid \mathbf{y}^R, \mathbf{x}^{-M})$ holds if and only if $\mathbf{Y}^i$ remains unchanged when tokens at positions $R$ are revealed.

**Stability of Unmasked Tokens.** Recent works Hu et al. (2025); Ma et al. (2025) have demonstrated that during iterative inference, the query, key, and value vectors ($\mathbf{Q}^{-M}$, $\mathbf{K}^{-M}$, $\mathbf{V}^{-M}$) for already unmasked tokens, remain stable and can be cached for computational efficiency.

**Why Independence Stability Holds.** Let us return to Assumption 3.3. First, we discuss padding end-of-sequence (EOS) tokens, which are used to fill the unused suffix reserved for an answer. By design, if there is an EOS token in $x^{-M}$ to the left of position $i$ then $p^i(\mathrm{EOS}|x^{-M}) = 1$ and the assumption automatically holds.

For regular tokens, we note that the stability property implies that in both cases, when we condition on $(\mathbf{y}^U, \mathbf{x}^{-M})$ or $(\mathbf{x}^{-M})$, the representations $(\mathbf{Q}^{-M}, \mathbf{K}^{-M}, \mathbf{V}^{-M})$ stay the same, while the main change happens for tokens in $U$.

The stability property allows us to concentrate on the information flow between position $i$ and tentatively unmasked subset $\mathbf{y}^U$, which we recall governed by attention weights vector $\mathbf{A}_{iU} := \left(A_{i,u_1}, \ldots, A_{i,u_{|U|}}\right)$. Specifically, we argue that a token $y_i$ is conditionally independent of a set of tokens $\mathbf{y}^U$ given the remaining tokens $\mathbf{x}^{-M}$ if and only if the total attention from position $i$ to all positions in $U$ is negligible across all layers and heads, or more formally, if $\|\mathbf{A}_{iU}\|_1 = \sum_{u \in U} A_{iu} < \delta$ for some small $\delta > 0$.

Now consider any subset $W \subset U$. The non-negativity of attention weights (a direct consequence of the softmax operation) yields the inequality:

$$\|\mathbf{A}_{iW}\|_1 = \sum_{w \in W} A_{iw} \leq \sum_{u \in U} A_{iu} = \|\mathbf{A}_{iU}\|_1 < \delta$$

This demonstrates that if position $i$ pays negligible attention to the entire set $U$, it necessarily pays negligible attention to any subset $W \subset U$. Consequently, the conditional distribution at position $i$ remains approximately invariant when conditioning on tokens at positions in $W$: $p_\theta^i(\cdot \mid \mathbf{y}^W, \mathbf{x}^{-M}) \approx p_\theta^i(\cdot \mid \mathbf{x}^{-M})$. This relationship directly corresponds to Assumption 3.3 (Independence Stability).

**Empirical Validation**    We use the same setup as was used in Section 4.3, as a source of prompts, we use the first round requests from MTBench, and sample the responses using the PUNT algorithm with different thresholds $\varepsilon = \{0.01, 0.04, 0.16, 0.64\}$.

For a step of the PUNT sampler with threshold $\varepsilon$, let $R$ denote the set of tokens unmasked at this step, sorted according to the confidence, $\mathbf{y}^R$ denotes the set of sampled candidates, and $x^{-M}$ denotes the set of already revealed tokens.

As we demonstrated at Fig. 3 sampled tokens satisfy

$$\delta_{\mathrm{KL}} = D_{\mathrm{KL}}\big(p^{r_i}(\cdot \mid \mathbf{x}^{-M}) \big\| p^{r_i}(\cdot \mid \mathbf{x}^{-M}, \mathbf{y}^{R_{<i}})\big) < \varepsilon.$$

For each token $r_i$, we compute the total attention from token $r_i$ to previously revealed tokens $R_{<i}$ for all heads of the last layer, i.e.

$$\delta_A = \|\mathbf{A}_{r_i R_{<i}}\|_1 = \sum_{j<i} A_{r_i r_j}$$

and plot (Fig. 6) the distribution of pairs $(\delta_{\mathrm{KL}}, \delta_A)$ for different thresholds.

We also compute the change of the layer output $\mathbf{Y}^{r_i}$ and how it changes when we condition on $\mathbf{y}^{R_{<i}}$. We use the normalized difference metric to compute the change, which is defined as normalized_difference$(a, b) := \|a - b\|_2 / \|a\|_2$, and plot (Fig. 7) the distribution of the change. Finally, similar to unmasked tokens, we observed that representations $\mathbf{Q}^{r_i}$, $\mathbf{K}^{r_i}$, $\mathbf{V}^{r_i}$ of masked token $r_i$ in the attention layer also stays stable when we additionally condition on previously revealed tokens $\mathbf{y}^{R_{<i}}$.

---

**Algorithm 1** PUNT (Parallel Unmasking with Non-influence Tests)

---

1: **Input:** masked sequence $\mathbf{x}$, vector of candidates $\mathbf{y}$, threshold $\varepsilon$
2: **Output:** certified set $R \subseteq M$ to unmask in parallel
3: Sort masked indices w.r.t. confidence heuristic $\phi$ in decreasing order
4: Construct $M$, the set of all masked indices.
5: $R \leftarrow M$
6: Let $B_b := \{i \in [|M|] : $ the $b$-th bit of $\mathrm{bin}(i) = 0\}$.
7: **for** $b$ in $[\log |M|]$ **do**
8:      $S_0 \leftarrow R \cap B_b$;                            (positions to tentatively unmask)
9:      $S_1 \leftarrow R \setminus B_b$;                           (positions to check for dependence)
10:      **for** each $j \in S_1$ **do**
11:          $d_j \leftarrow D_{\mathrm{KL}}\big(p^j(\cdot \mid \mathbf{x}^{-M}) \big\| p^j(\cdot \mid \mathbf{x}^{-M}, \mathbf{y}^{S_1})\big)$
12:          **if** $d_j > \varepsilon$ **then**
13:             $R \leftarrow R \setminus \{j\}$
14:          **end if**
15:      **end for**
16: **end for**
17: **return** $R$

---

## C    IMPLEMENTATION AND EXPERIMENTS

This section evaluates the proposed planner PUNT (Algorithm 1) across diverse sequence generation tasks. All experiments are conducted on A100 GPUs with 40GB memory.

| Experiment | NumFewshot | max length |
|------------|------------|-----------:|
| GSM8K | 4 | 512 |
| HumanEval | 0 | 512 |
| MBPP | 3 | 512 |
| IFEval | 0 | 1024 |
| MT-BENCH | - | 1024 |

Table 1: Experimental configuration for each benchmark task.

PUNT offers a clear win in step efficiency without compromising on quality. However, this is not indicative of the underlying compute used, which is better captured by the number of forward passes (NFE). In terms of NFE, it performs competitively, and particularly on long-sequence tasks, it often surpasses the baselines. We leave further per-step optimisation for future work.

## C.1 EXPERIMENTAL SETUP

We evaluate two state-of-the-art discrete diffusion models for natural language: **LLaDA-1.5** (Zhu et al., 2025a) and **Dream-v0-Instruct-7B** (Ye et al., 2025a) (referred to as Llada and Dream, respectively). In this section, we detail the experimental setup, including tasks, datasets, evaluation metrics, and baseline methods.

### TASKS AND DATASETS

We assess PUNT's performance on a variety of sequence generation tasks. The evaluation relies on the following standard public datasets and their corresponding protocols:

- Math word problems and formal math: GSM8K (Cobbe et al., 2021), MATH (Hendrycks et al., 2021)
- Code generation: HumanEval (Chen et al., 2021) and MBPP (Austin et al., 2021).
- Instruction-following evaluation: IFEval (Zhou et al., 2023)
- Open-ended question benchmarks: MT-Bench (Zheng et al., 2023).

### EVALUATION METRICS AND CONFIGURATION

We use task-specific evaluation metrics and measure efficiency in terms of the number of forward evaluations and the number of iterations PUNT takes.

**Quality Metrics:**

- Math problems: Match accuracy (GSM8K)
- Code generation: Pass@1 success rate (HumanEval, MBPP)
- Instruction following: Strict/Loose prompt/instruction adherence (IFEval)
- Open-ended generation: GPT-4o scoring 1-10 (MT-Bench)

**Efficiency Metrics:**

- Number of network function evaluations (NFE) per sequence
- Number of generation steps (PUNT-specific)

### BASELINE METHODS

We compare against representative training-free schedulers with the following parameters:

- Top-k Sampler with $k = 1, 2, 3, 4, 5, 6$;
- EB-Sampler (entropy-bounded unmasking) with $\epsilon = 0.01, 0.05, 0.1, 0.5, 1.0, 2.0, 4.0$ (Patel et al., 2025);

- Geometry-aware spacing: dilated with log window size in $\{3, 4, 5, 6, 7\}$ (Luxembourg et al., 2025),

Each of these baselines utilizes a confidence score to rank positions by certainty. Different options for the confidence score are described below.

## CONFIDENCE SCORING STRATEGIES

All confidence scoring strategies operate on the model's output probability distribution. For each position $t$ in a sequence, the model produces logits $l_{b,t,v}$ for every token $v$ in the vocabulary. These are converted into a probability distribution using the softmax function:

$$p_{b,t,v} = \frac{e^{l_{b,t,v}}}{\sum_{v'=1}^{V} e^{l_{b,t,v'}}}.$$

From this distribution, we compute a scalar confidence score $s_{b,t}$ that quantifies the model's certainty at that position. A higher score indicates greater confidence, prioritizing that position for earlier unmasking. To define the scoring strategies, we use the following notation:

- $p_{b,t,(k)}$: The $k$-th largest probability at position $t$, such that $p_{b,t,(1)} \geq p_{b,t,(2)} \geq \cdots \geq p_{b,t,(V)}$.
- $y_{b,t}$: The token actually sampled at position $t$.

**Negative Entropy**

$$s_{b,t} = \sum_{v=1}^{V} p_{b,t,v} \log p_{b,t,v} = -H(p_{b,t}), \qquad H(p_{b,t}) = -\sum_{v=1}^{V} p_{b,t,v} \log p_{b,t,v}.$$

This is the *negative* Shannon entropy. Values lie in $\left[-\log V, 0\right]$. Scores closer to $0$ correspond to more peaked (certain) distributions.

**Top Probability**

$$s_{b,t} = \max_v p_{b,t,v} = p_{b,t,(1)}.$$

A simple peak-confidence heuristic. Ignores how close competitors are.

**Top Probability Margin**

$$s_{b,t} = p_{b,t,(1)} - p_{b,t,(2)}.$$

Measures local ambiguity between the two most likely tokens. Larger margin $\Rightarrow$ clearer preference.

**Positional Schedule**

$$s_{b,t} = t.$$

A deterministic curriculum ignoring model uncertainty (e.g. left-to-right). Negate or reverse indices if the opposite order is desired.

## C.2 IMPLEMENTATION DETAILS

Our implementation of PUNT follows the procedure outlined in Algorithm 1. To ensure a fair comparison, both PUNT and the baseline methods use the same confidence scoring strategy for each model. Specifically, we use the top probability margin for LLaDA and negative entropy for Dream.

### SAMPLING AND TEMPERATURE SETTINGS

All methods employ nucleus sampling with nucleus mass set to $0.9$. We present results for two temperature settings: $0.1$ (low temperature, focused sampling) and $0.7$ (higher temperature, more diverse sampling) to evaluate robustness across different generation regimes.

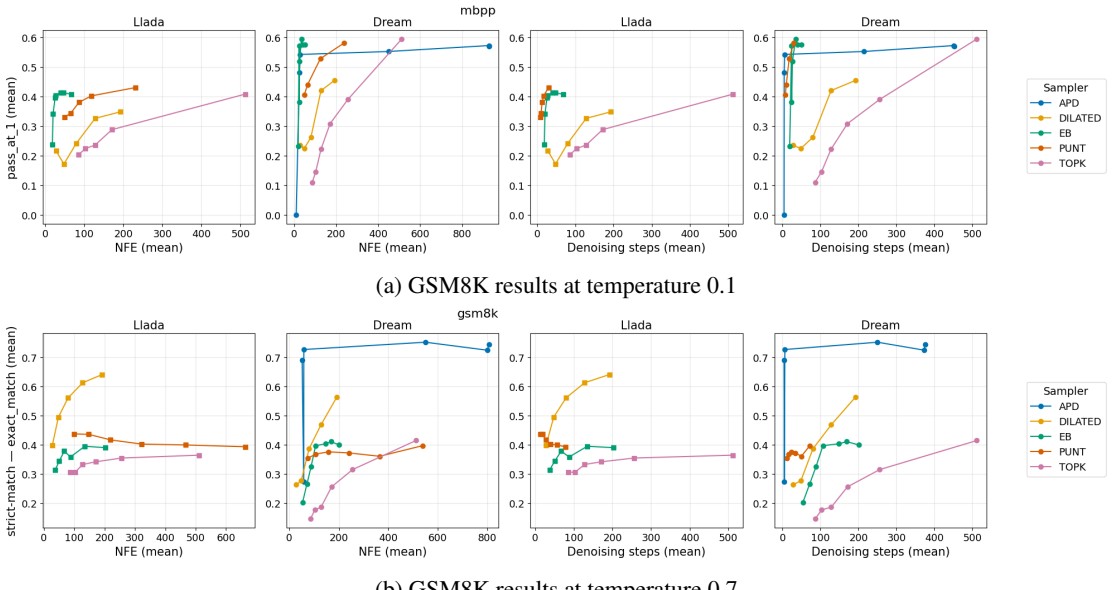

(a) GSM8K results at temperature 0.1

(b) GSM8K results at temperature 0.7

Figure 8: GSM8K performance comparison across different temperature settings showing NFE/steps vs match accuracy (flexible-extract filter)

END-OF-SEQUENCE HANDLING

To prevent premature termination, we down-weight positions corresponding to end-of-sequence tokens when early termination is undesirable: If $y_{b,t}$ equals a special end-of-sequence token EOS and early termination is undesirable, enforce

$$s_{b,t} \leftarrow C_{\text{neg}}, \qquad C_{\text{neg}} \ll 0,$$

to deprioritize revealing that position.

C.3    RESULTS AND ANALYSIS

We present our results grouped by sequence length, as this factor significantly impacts the relative performance of the scheduling methods.

SHORT-SEQUENCE BENCHMARKS

We evaluate PUNT on GSM8K, HumanEval, and MBPP —all tasks with sequences shorter than 1024 tokens (see Table 1). These benchmarks test mathematical reasoning and code generation capabilities under constrained generation lengths.

**Results:** When measured by the number of generation steps, PUNT consistently outperforms all baseline methods across both temperature settings (0.1 and 0.7). However, when evaluated by NFEs per sequence, PUNT shows competitive but not dominant performance. PUNT's strength lies in reducing the number of sequential generation steps through aggressive parallelization, but each step may require more network evaluations due to its comprehensive independence testing.

LONG-SEQUENCE BENCHMARKS

For longer sequences ($\geq$ 1024 tokens), we evaluate on MT-Bench and IFEval. These tasks require sustained coherence and complex instruction following over extended generation windows.

**MT-Bench Results:** MT-Bench consists of open-ended questions spanning creative writing, reasoning, and coding. Each question includes two rounds, where the second builds upon the first response. Answers are evaluated by GPT-4o using a 1-10 scale. All experiments are carried out with temperature 0.7.

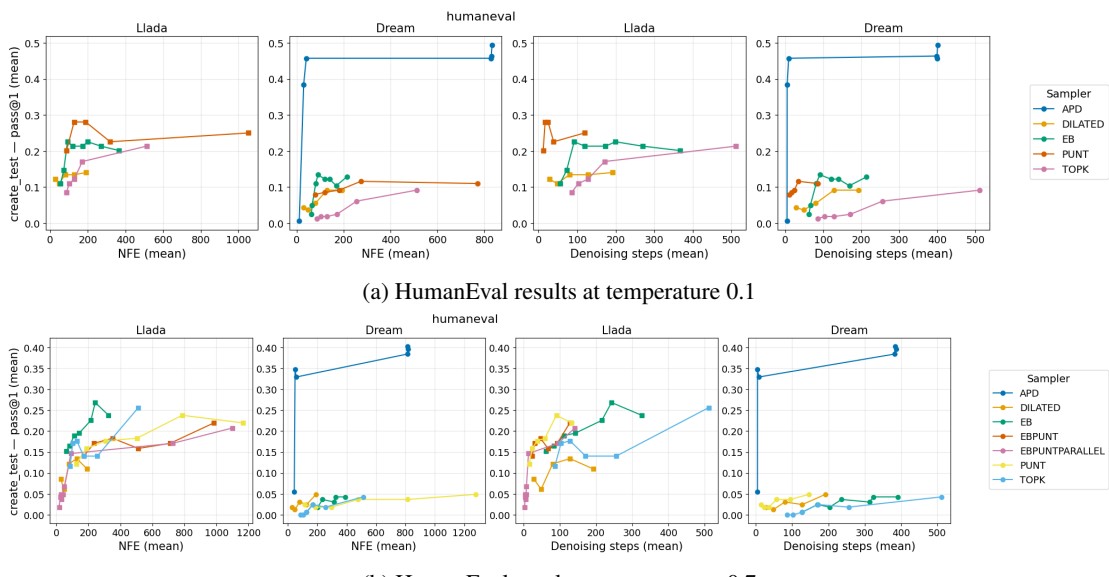

(a) HumanEval results at temperature 0.1

(b) HumanEval results at temperature 0.7

Figure 9: HumanEval performance comparison across different temperature settings showing NFE/steps vs Pass@1 success rate for both LLaDA and Dream models

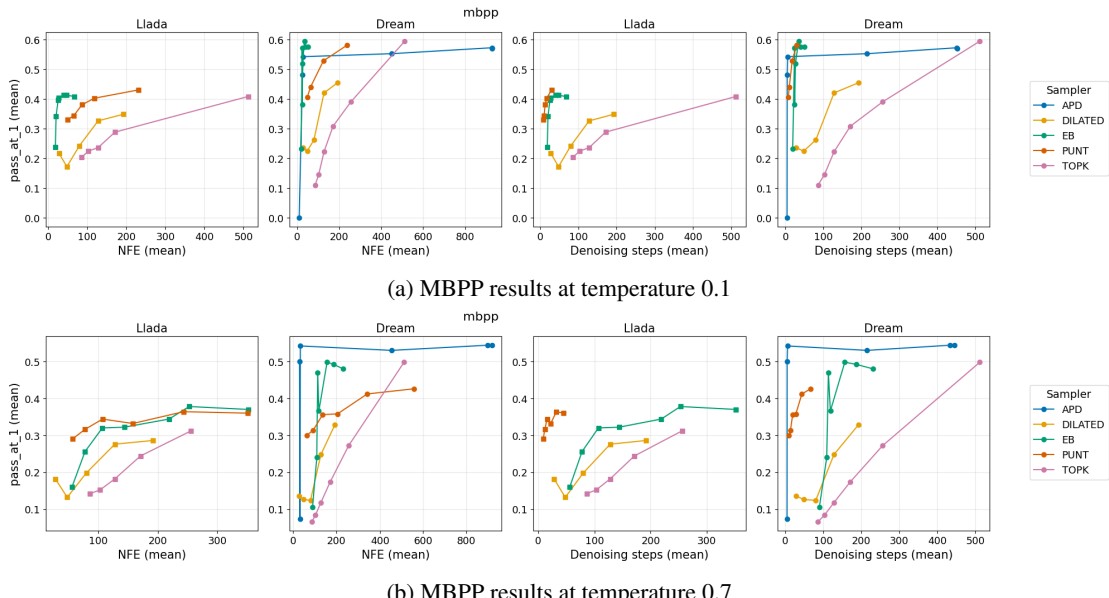

(a) MBPP results at temperature 0.1

(b) MBPP results at temperature 0.7

Figure 10: MBPP performance comparison across different temperature settings showing NFE/steps vs Pass@1 success rate for both LLaDA and Dream models

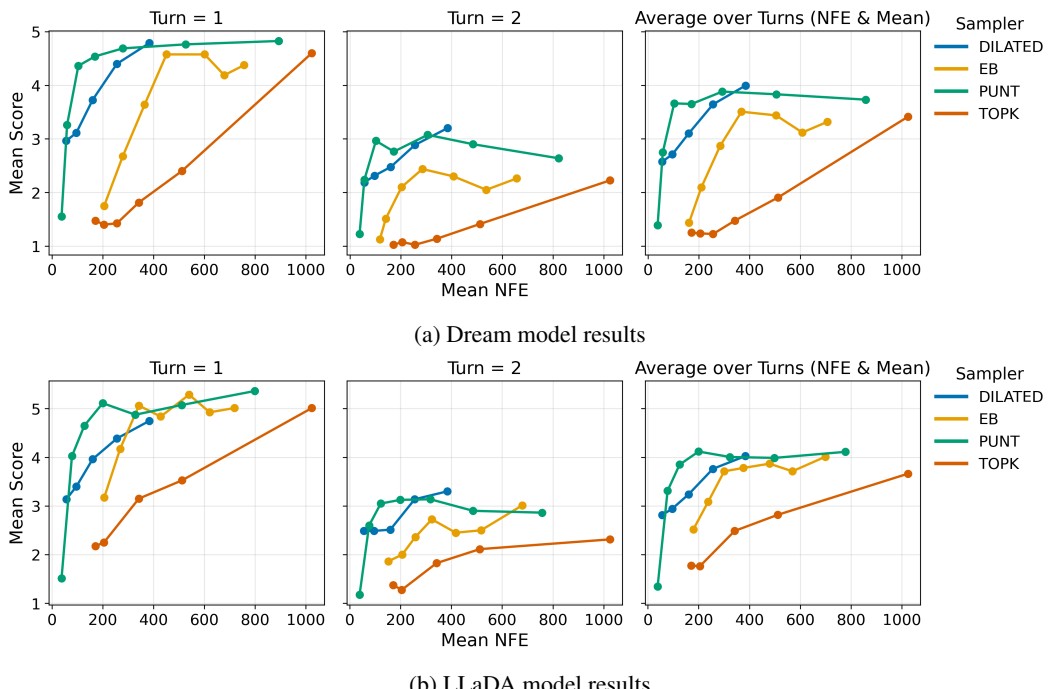

(a) Dream model results

(b) LLaDA model results

Figure 11: Performance comparison for MT-Bench across different models showing NFE vs mean performance

Figure 11 show that PUNT excels particularly when NFE budgets are severely constrained. In low-NFE regimes, PUNT significantly outperforms all baseline methods. As the NFE budget increases, dilated sampling begins to show competitive performance, but PUNT maintains its characteristic stability advantage.

**IFEval Results:**

The instruction-following evaluation tests adherence to specific formatting and content constraints. PUNT demonstrates consistent accuracy across both NFE and step-based metrics, again showing its reliability advantage.

PUNT's results on IFEval demonstrate its stability across different computational budgets. As shown in Figures 12 and 13, it consistently leads in generation steps at both temperatures, without compromising accuracy. Additionally, PUNT is more NFE-efficient at lower budgets and remains competitive as the budget increases, pulling ahead at a temperature of 0.7.

## C.4 APD EXPERIMENTS

We evaluate the performance of PUNT against Adaptive Parallel Decoding (APD) proposed in (Israel et al. (2025)). APD runs have been added to all benchmarks running on Dream (Ye et al., 2025a). There are no comparisons on benchmarks running on Llada (Zhu et al., 2025a). APD requires a trained autoregressive model using the same tokenizer as the diffusion model, but no such compatible autoregressive model exists, as noted in (Israel et al., 2025).

APD requires two NFEs per sampling step: one to draw logits from the diffusion model; a second full decoding pass from the autoregressive model, to build the target distribution. Both distributions are sampled using Gumbel-Softmax trick, and tokens are accepted when the samples coincide in both processes, using a left-to-right decoding scheme. When plotting APD's accuracy vs NFEs, we multiplied by 2 the denoising steps used by APD to obtain NFEs: one for the diffusion model and another to construct the target distribution from the autoregressive model.

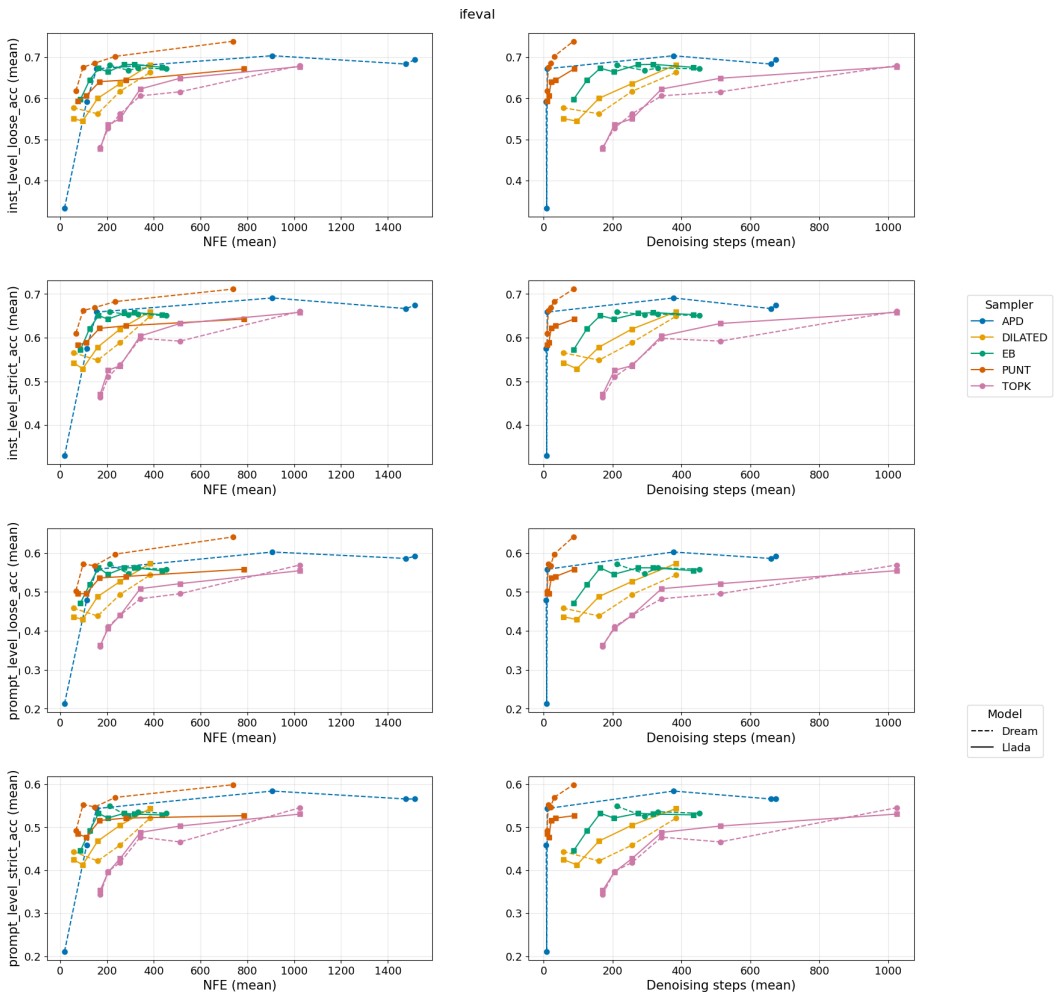

Figure 12: IFEval results showing NFE/steps vs accuracy, temperature 0.1.

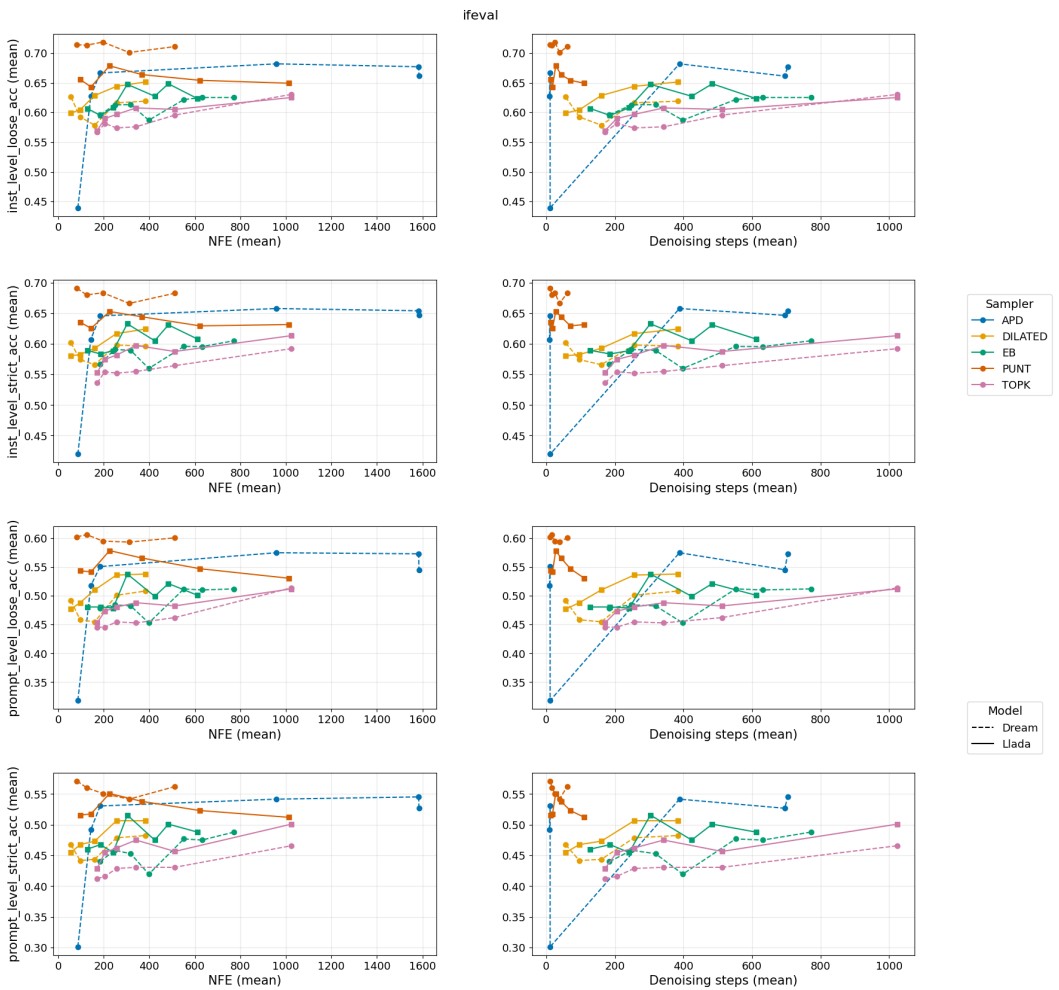

Figure 13: IFEval results showing NFE/steps vs accuracy, temperature 0.7.

All experiments discussed below refer to the Dream architecture. As illustrated in Figs. 12 and 13, PUNT demonstrates superior performance on IFEval, achieving the highest scores among all methods. This indicates a strong capability in adhering to complex constraints and instructions. APD shows the strongest performance in HumanEval Fig. 9a and Fig. 9b. EB and PUNT surpass APD on MBPP Fig. 10a at temperature 0.1, and APD surpasses them at temperature 0.7. EB surpasses APD and PUNT on GSM8K at temperature 0.1, see Fig. 8a, and APD surpasses all other samplers at temperature 0.7, see Fig. 8b.

Dream is a diffusion model whose weights have been initialized from a trained autoregressive model (Qwen et al., 2025). We hypothesize that Dream retains good left-to-right sampling performance because of its weight initialization, and APD's performance benefits because of its strict left-to-right sampling order when used with Dream. In contrast, all other samplers operate independently of the diffusion model's training procedure, allowing them to be applied more broadly.

### C.5 MASKED DIFFUSION MODELS FOR PROTEINS

Masked diffusion models (MDMs) have demonstrated effectiveness beyond natural language processing, particularly in generating biological sequences such as proteins and DNA. To evaluate PUNT's performance in a structured biological domain, we conduct experiments on *de novo* membrane protein design using MemDLM (Goel et al., 2024), a masked diffusion model that finetunes the state-of-the-art ESM-2 150M protein language model (Lin et al., 2023) with an MDM objective to generate realistic membrane proteins.

### C.5.1 EXPERIMENTAL SETUP

We evaluate PUNT on unconditional protein generation with sequences of up to 1024 amino acids, comparing against three established training-free schedulers: Top-$k$ sampling, Entropy-Bound (EB) unmasking, and geometry-aware (Dilated) spacing. All methods employ a temperature of 0.8, to encourage sequence novelty, and suppress end-of-sequence tokens to promote longer, more realistic protein sequences. For each sampling strategy, we generate 50 amino acid sequences using the following hyperparameters:

- PUNT: $\varepsilon = \{0.001, 0.004, 0.01, 0.02, 0.04, 0.08, 0.16\}$
- Top-$k$: $k = \{1, 2, 3, 4, 6, 8, 12\}$
- EB Sampler: $\varepsilon = \{0.1, 0.5, 1, 5, 10\}$
- Geometry-aware spacing: $\log w = \{3, 4, 5, 6, 7, 10\}$

### C.5.2 EVALUATION METRICS

We assess PUNT's performance across two key dimensions critical for practical protein design applications (Wenran et al., 2025):

**Computational Efficiency:** As with the natural language benchmarks, we measure the number of forward evaluations (NFE) and denoising steps required for generation. NFE represents the total number of model forward passes needed to complete sequence generation, providing a direct measure of computational cost. Denoising steps (PUNT-specific) track the number of iterative refinement steps in the masked diffusion process.

**Structural Validity:** Generated protein sequences are evaluated for their likelihood to fold into stable, well-defined three-dimensional structures. We feed each generated amino acid sequence to ESMFold (Lin et al., 2023) to predict the corresponding 3D protein structure. We then calculate the mean pLDDT (a per-residue measure of local confidence in the structural predictions) across all residues in each predicted structure.

### C.5.3 RESULTS AND ANALYSIS

Figure 14 plots the mean pLDDT against NFE and number of denoising steps. We find that while the pLDDT of generated structures is low across denoising methods—which may be attributed in part to

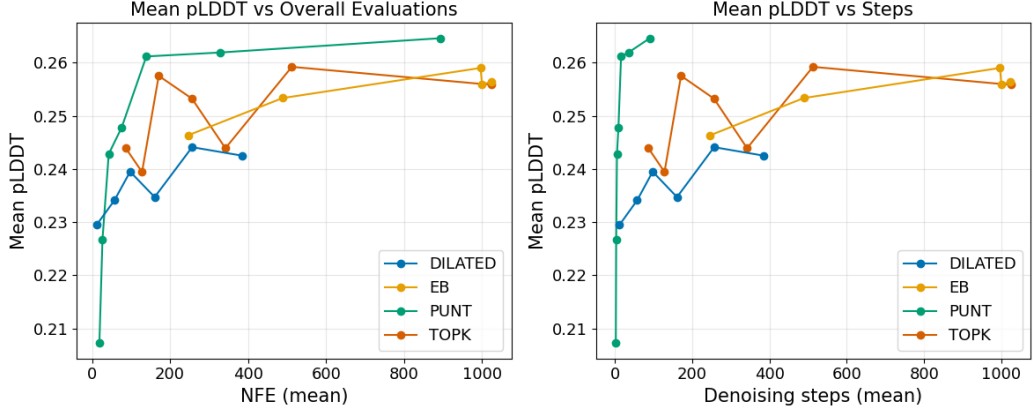

Figure 14: Protein generation with MeMDLM: mean pLDDT vs (a) NFE, and (b) denoising steps.

our use of a non-semi-autoregressive generation strategy, or because of the very long sequence length and absence of multiple sequence alignment in ESMfold making this a challenging domain for 3D structure prediction—PUNT consistently generates proteins with comparable or marginally higher pLDDT than the baseline samplers given the same computational budget, with stable performance across a broad range of NFE. These results suggest that PUNT is able to improve efficiency without sacrificing structural plausibility, making it well-suited for rapid proposal of candidate proteins for downstream analysis.

## D  DENOISING PROCESS

In this section, we show examples of our denoising process starting from a completely masked response for three prompts from different domains.

### TEXT PROMPT

The first prompt is a story generation prompt: *"Compose an engaging story about a recent trip to Hawaii, highlighting cultural experiences and must-see attractions."*

### MATH PROMPT

The second prompt is a math word problem: *"Natalia sold clips to 48 of her friends in April, and then she sold half as many clips in May. How many clips did Natalia sell in altogether in April and May?"*

### PROGRAMMING PROMPT

The third prompt is a programming prompt: *"Write a function to find the minimum cost path to reach $(m, n)$ from $(0, 0)$ for the given cost matrix cost[][] and a position $(m, n)$ in cost[][]."*

## E  PLOTTING PERFORMANCE VS $\epsilon$

In this Figures 15 to 18, we plot the performance of different models as a function of the closeness parameter $\epsilon$ for two different temperature settings: 0.1 and 0.7.

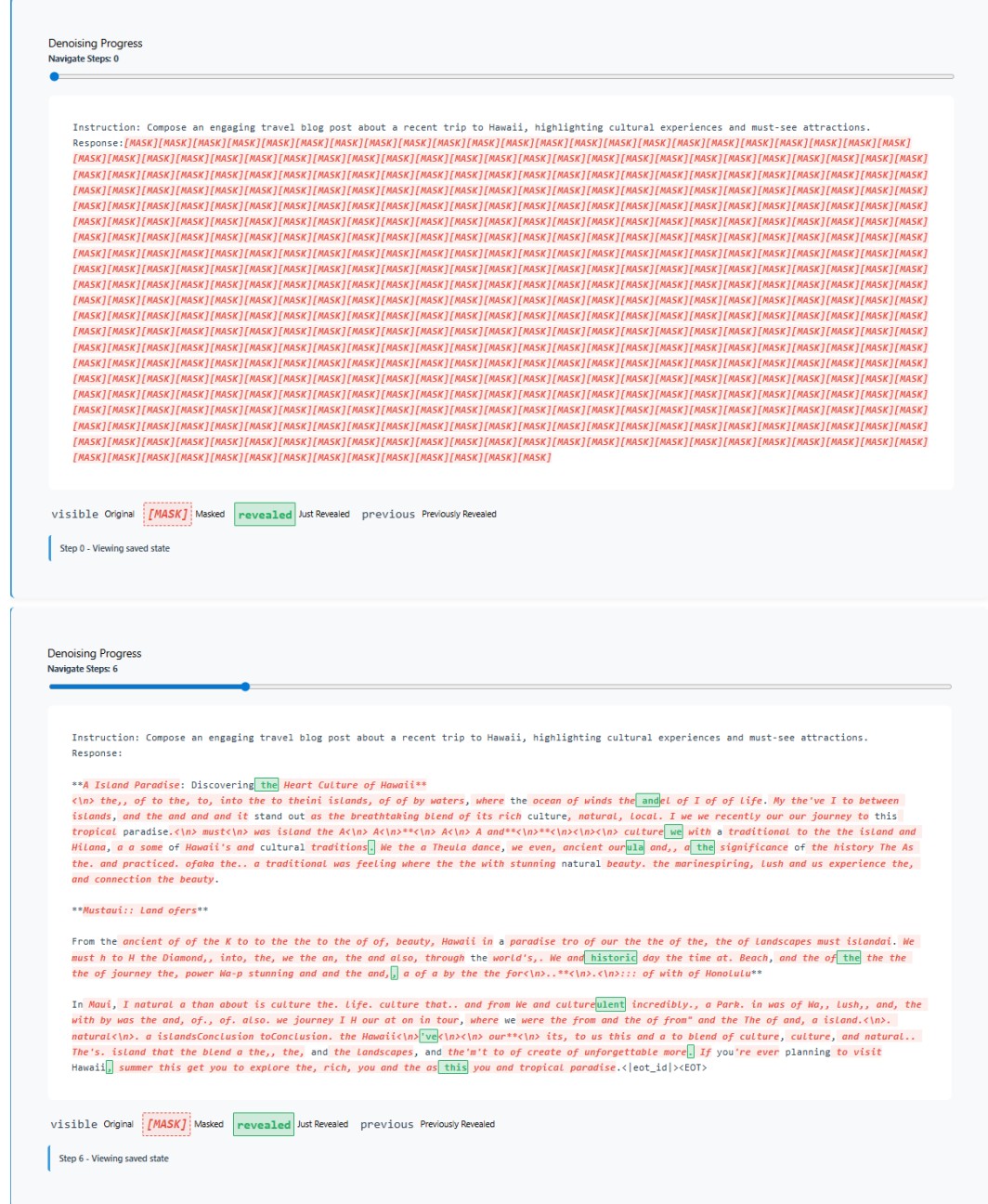

## F    PERFORMANCE FOR VARYING EPSILON SCHEDULES

In this section, we present the performance of PUNT with varying epsilon schedules compared to the pruning baseline on the MBPP and IFEval benchmarks. The results are shown in Figures 19, 20, and 21.

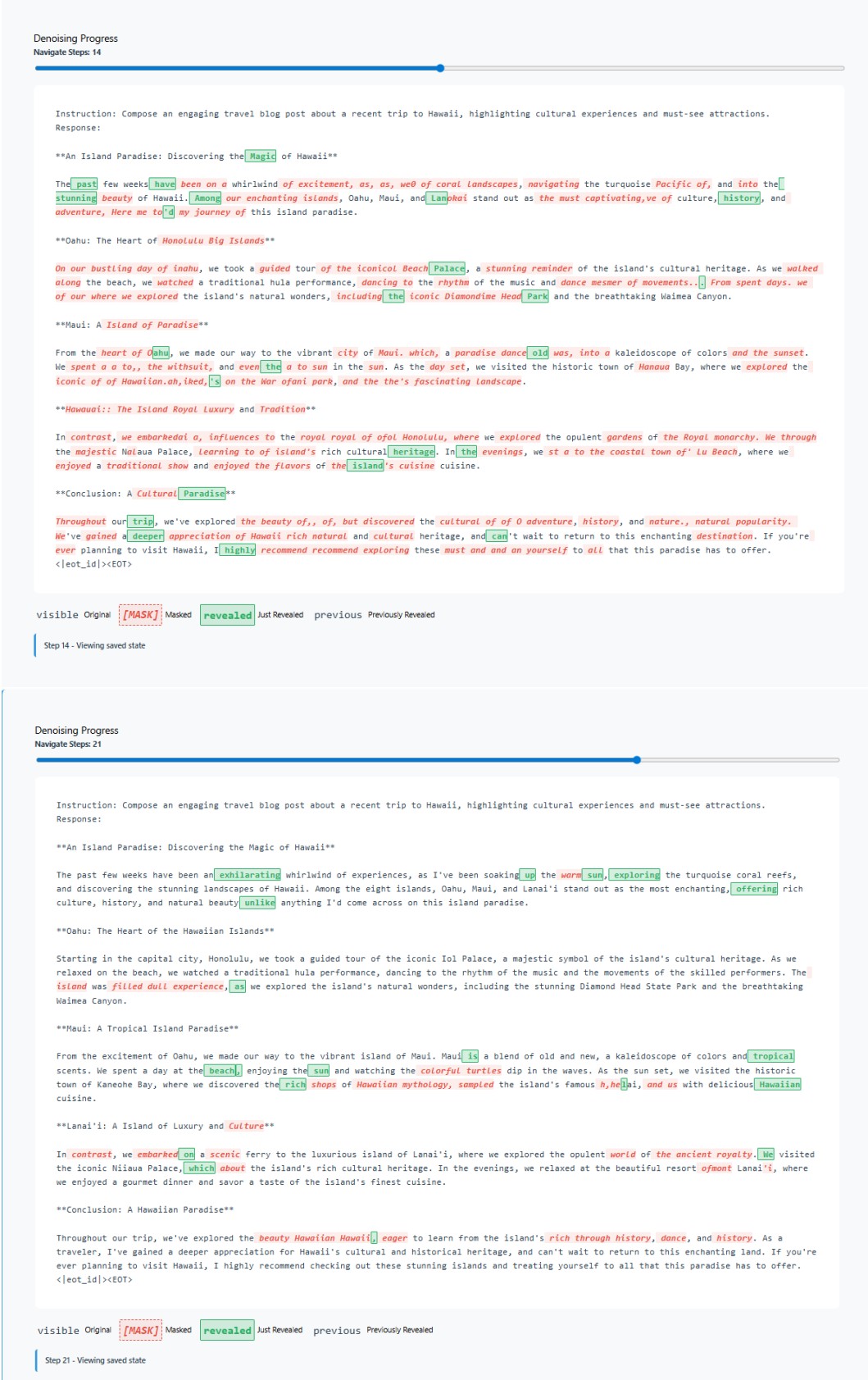

Denoising Progress
Navigate Steps: 28

Instruction: Compose an engaging travel blog post about a recent trip to Hawaii, highlighting cultural experiences and must-see attractions.
Response:

An Island Paradise: Discovering the Magic of Hawaii

The past few weeks have been an exhilarating whirlwind of experiences, as I've been soaking up the tropical sun, exploring the turquoise coral reefs, and discovering the stunning landscapes of Hawaii. Among the eight islands, Oahu, Maui, and Lanai'i stand out as the most enchanting, offering rich culture, history, and natural beauty unlike anything I'd come across on this island paradise.

Oahu: The Heart of the Hawaiian Islands

Starting in the capital city, Honolulu, we took a guided tour of the iconic Iol Palace, a majestic symbol of the island's cultural heritage. As we relaxed on the beach, we watched a traditional hula performance, dancing to the rhythm of the music and the movements of the skilled performers. The day was filled with adventure, as we explored the island's natural wonders, including the stunning Diamond Head State Park and the breathtaking Waimea Canyon.

Maui: A Tropical Island Paradise

From the excitement of Oahu, we made our way to the vibrant island of Maui. Maui is a blend of old and new, a kaleidoscope of colors and tropical scents. We spent a day at the beach, enjoying the sun and watching the dolphins gracefully dip in the waves. As the sun set, we visited the historic town of Kaneohe Bay, where we discovered the rich culture of the island, including the island's famous Luau 'lai, a feast with delicious Hawaiian cuisine.

Lanai'i: A Island of Luxury and Beauty

In Maui, we embarked on a scenic ferry to the luxurious island of Lanai'i, where we explored the opulent homes of the royal family. We visited the iconic Niiaua Palace, which showcases the island's rich cultural heritage. In the evenings, we relaxed at the beautiful resort of Royal Lanai'i, where we enjoyed a gourmet dinner and savor a taste of the island's finest cuisine.

Conclusion: A Hawaiian Paradise

Throughout our trip, we've explored the beauty of Hawaii, eager to learn from the island's rich cultural heritage, music, and traditions. As a traveler, I've gained a deeper appreciation for Hawaii's cultural and historical heritage, and can't wait to return to this enchanting land. If you're ever planning to visit Hawaii, I highly recommend checking out these stunning islands and treating yourself to all that this paradise has to offer.
<|eot_id|>

visible Original  [MASK] Masked  revealed Just Revealed  previous Previously Revealed

Step 28 - Viewing saved state

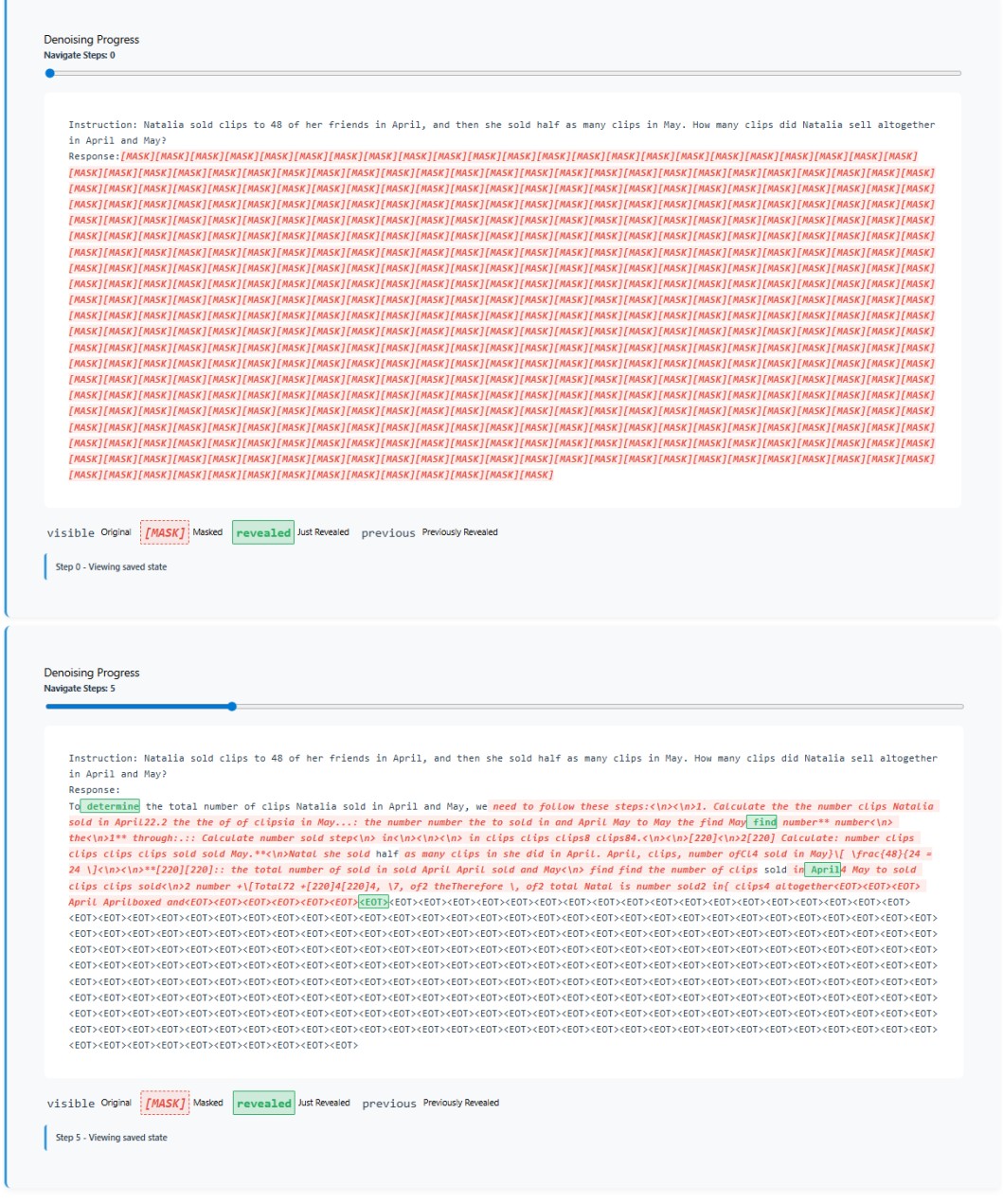

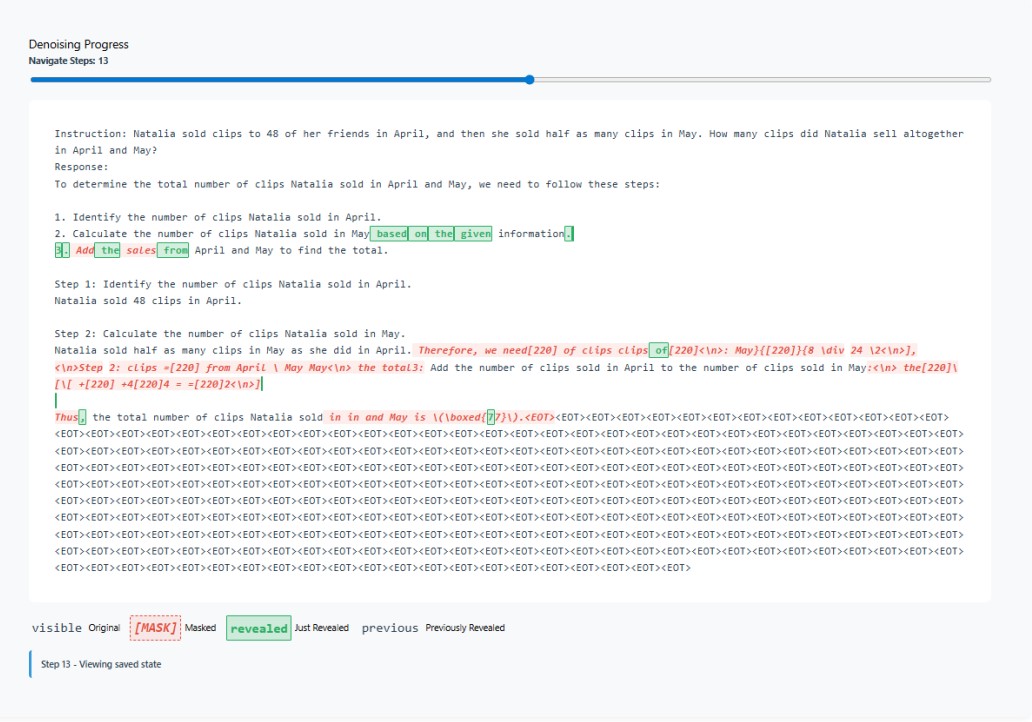

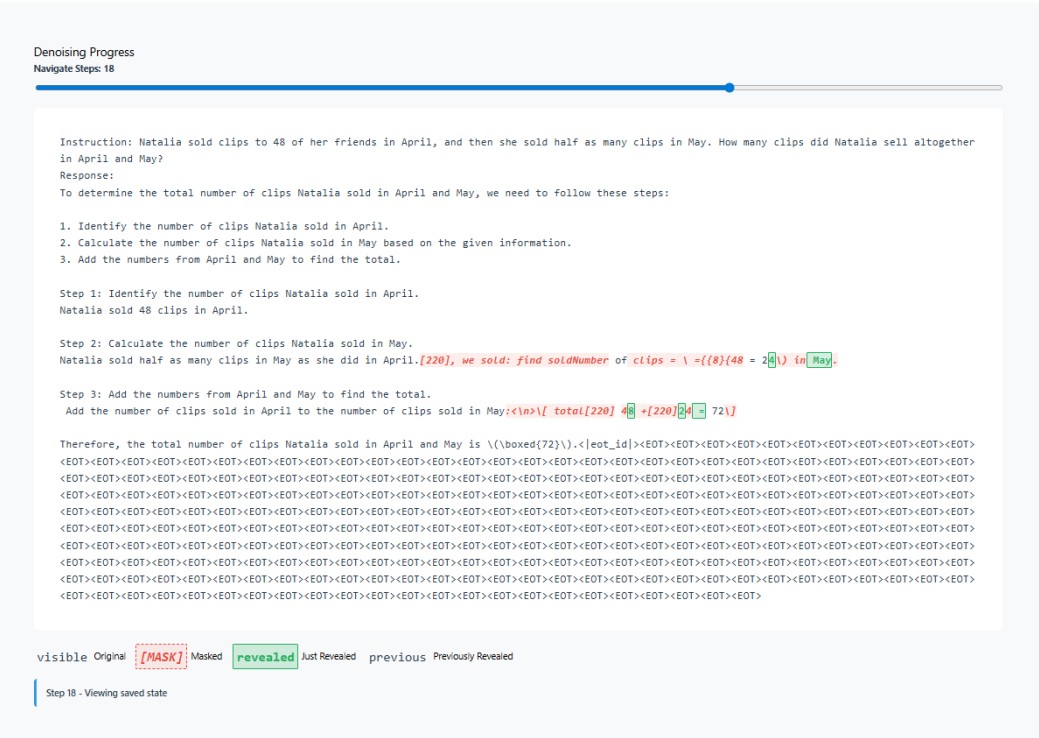

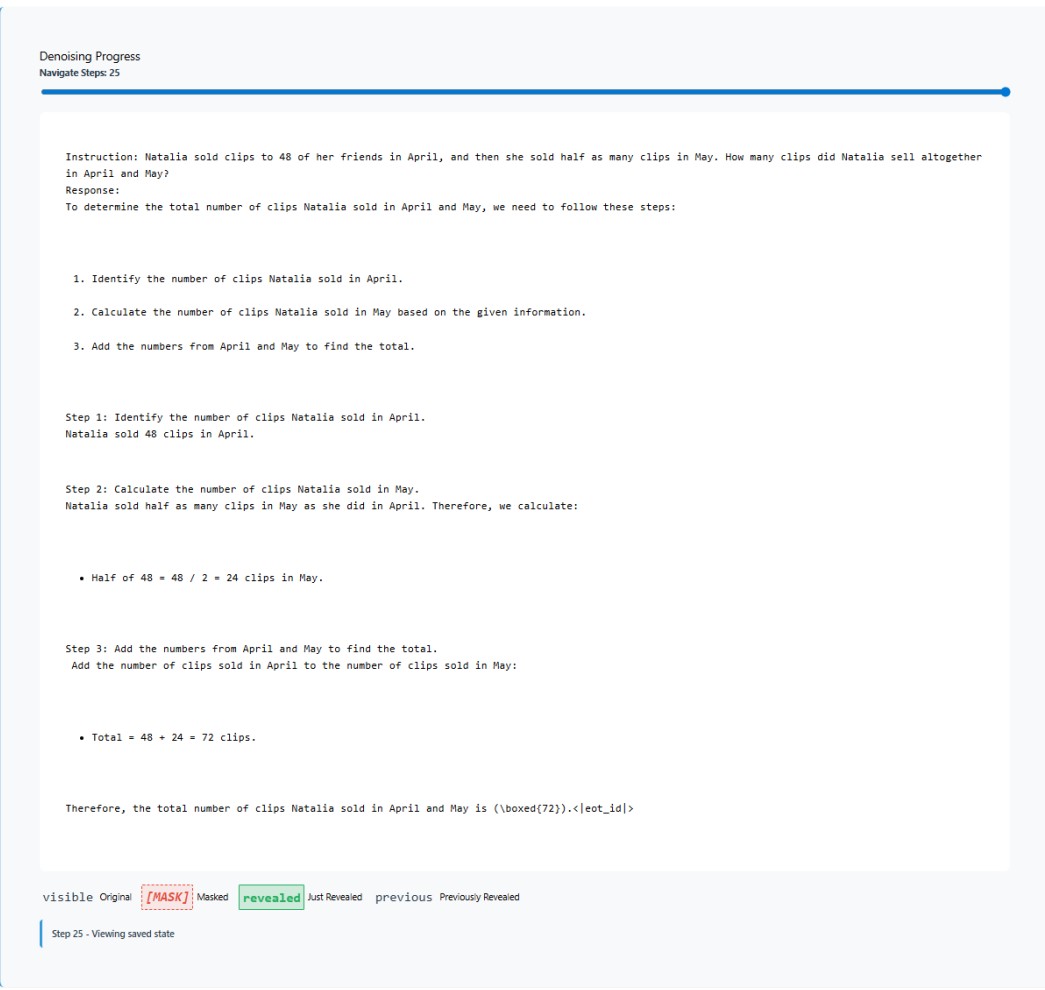

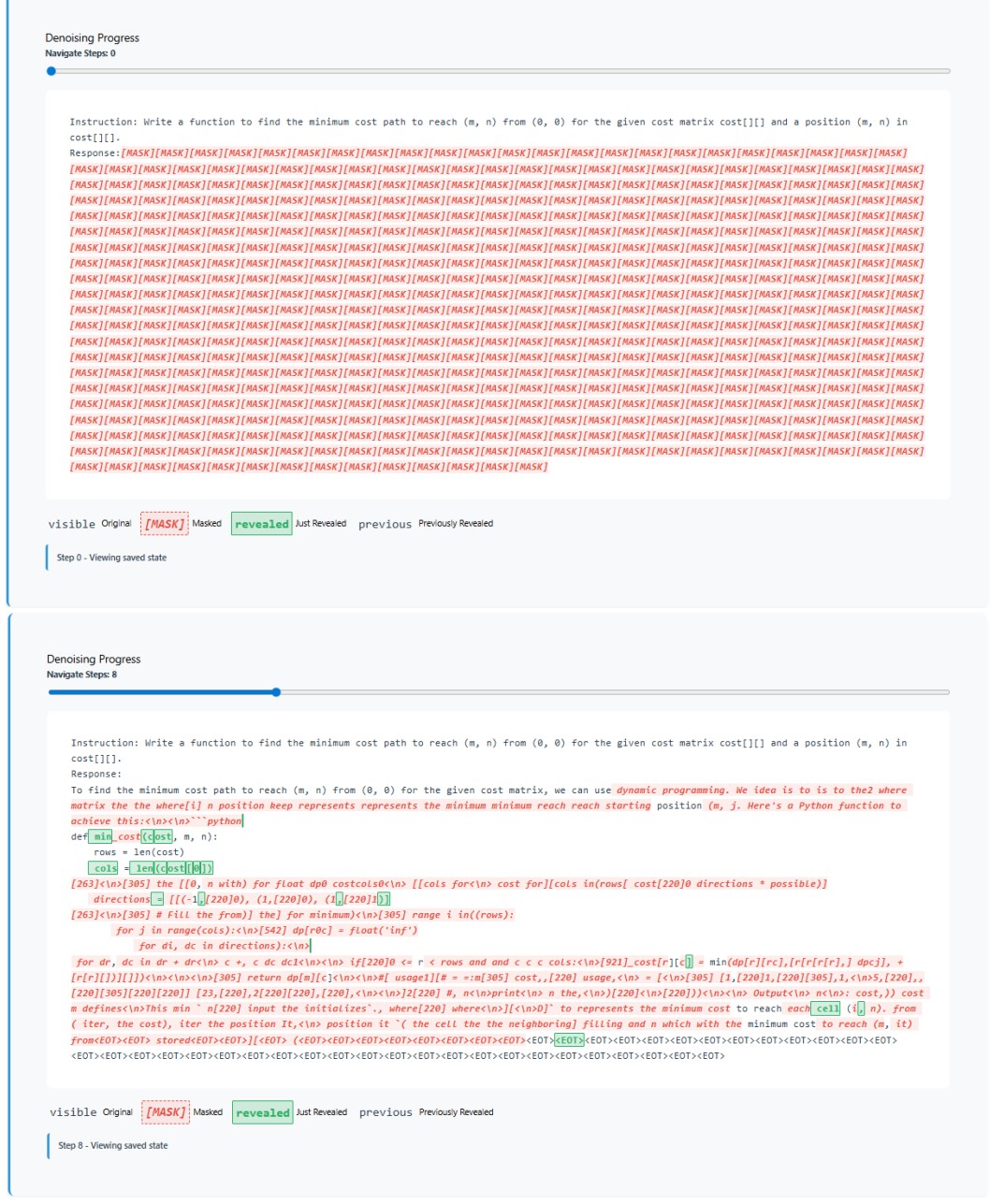

**Denoising Progress**
Navigate Steps: 16

Instruction: Write a function to find the minimum cost path to reach (m, n) from (0, 0) for the given cost matrix cost[][] and a position (m, n) in cost[][].
Response:
To find the minimum cost path to reach (m, n) from (0, 0) for the given cost matrix, we can use dynamic programming. The idea is to create a[220]2 a array cell costdp positionj,,) represents0 reach,[220]) reach reach position (i, j). Here is a Python function to achieve this:

```python
def min_cost(cost, m, n):
    rows = len(cost)
    cols = len(cost[0])
[305] min = [[min0, cols0 =[220]0] *[220] incols range<\n>]
[263] min[305] # Directions: (up, up,,,,)
    directions = [(-1, 0), (1, 0), (0, 1)]

    # Fill the cost array with with minimum up[263]
    for i in range(rows):
        for j in range(cols):
            dp[i][j] = float('inf')
            for dr, ddc in directions:
                r, c = i(i) + dr, j* j + dr<\n>[921] if[220]0 <= r < rows and[220]0 <= c < cols and cost[i[r][c] = min(dp[ijj<\n> min min[][jj min
dp[r][c] dp min[[rjc]
[263]
    return dp[m][n]

# Example usage:
cost_matrix = [
    [2, 3, 4],
    [6, 5, 1],
    [4, 2, 1]
]
m_m, end_n = (2,[220]2)<\n>print(min_cost(cost_matrix, m_pos, n_col))[220] # Output: 7
```

This function initializes initializes thedp matrix with the cost matrix and the the the directions. moving it iterates in the cost matrix. It, it calculate the minimum cost to reach each cell (i, j) by considering all over the the (,, right, down, right) and updating the `dp` matrix accordingly. Finally, it returns the minimum cost to reach (m, n) from (0, 0).<|eot_id|>

visible Original [MASK] Masked revealed Just Revealed previous Previously Revealed

Step 16 - Viewing saved state

---

**Denoising Progress**
Navigate Steps: 24

Instruction: Write a function to find the minimum cost path to reach (m, n) from (0, 0) for the given cost matrix cost[][] and a position (m, n) in cost[][].
Response:
To find the minimum cost path to reach (m, n) from (0, 0) for the given cost matrix, we can use dynamic programming. The idea is to build a matrix where each cell min[i][j] represents the minimum cost to reach that cell from the current position (i, j). Here is a Python function to achieve this:

```python
def min_cost(cost, m, n):
    rows = len(cost)
    cols = len(cost[0])
    min = [[ [(0, 0) in0] for _ in range(cols)]
    min = 3 possible4 move (up, down, right)
    directions = [(-1, 0), (1, 0), (0, 1)]

    # Calculate the minimum cost path to reach each position
    for i in range(rows):
        for j in range(cols):
            min[i][j] = float('inf')
            for dr, dc in directions:
                r, c = row[i] + dr, min[j] + dc
                if (0 <= r < rows and 0 <= c < cols) and min[r][c] < min[i][j]:
                    min[i][j] = min[r][c] + costdr][j]

    return min[m][n]

# Example usage:
cost_matrix = [
    [1, 3, 4],
    [6, 5, 1],
    [4, 2, 1]
]
end_row, end_col = 2, 2

print(min_cost(cost_matrix, end_row, end_col))  # Output: 6
```

This function initializes by ` matrix matrix matrix with to costs and the the possible directions. It then iterates through the cost matrix and min matrix, calculating the minimum cost to reach each cell (i, j) by considering the cost the adjacent cells (up, down, and right) and updating the minimum cost if they exist. Finally, it returns the minimum cost to reach (m, n) from (0, 0).<|eot_id|>

visible Original [MASK] Masked revealed Just Revealed previous Previously Revealed

Step 24 - Viewing saved state

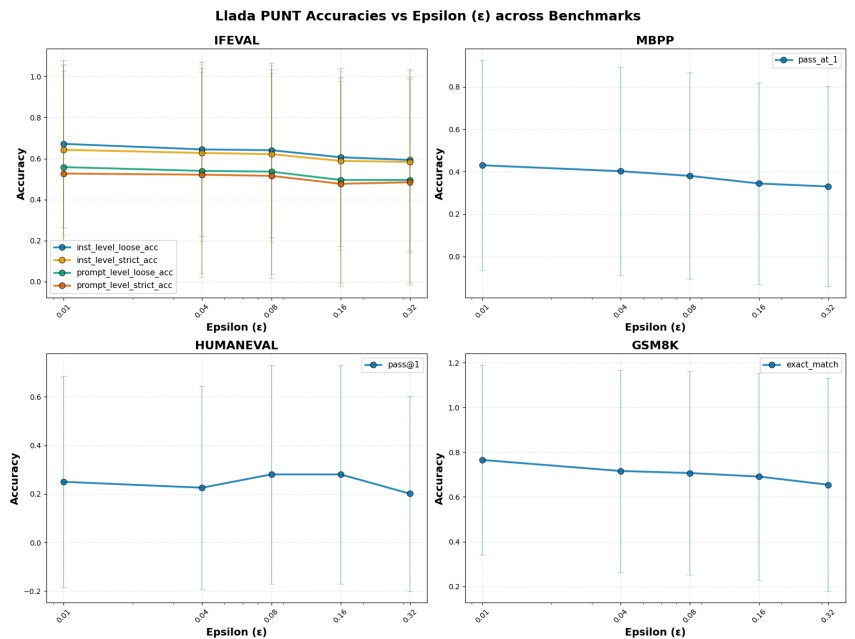

Figure 15: Performance vs $\epsilon$ for LLADA at temperature 0.1

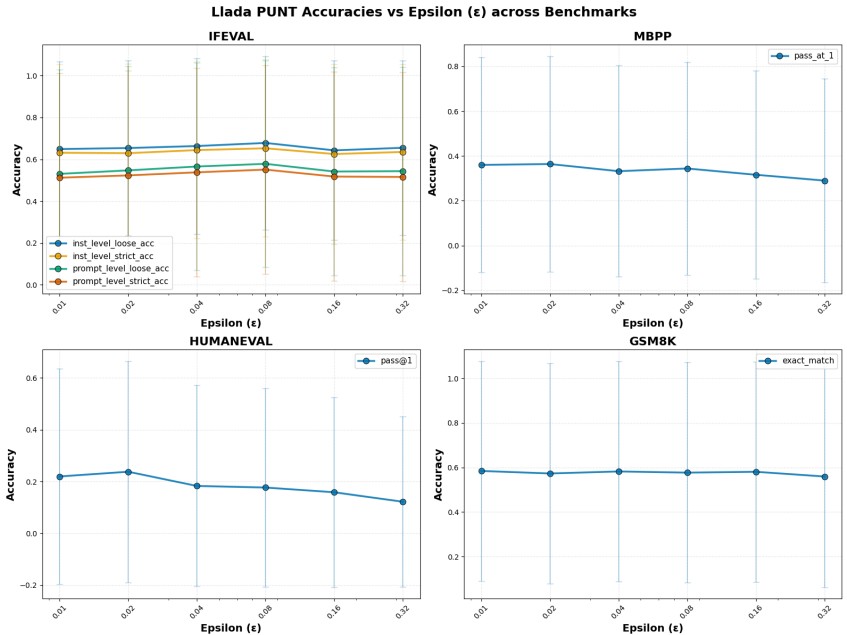

Figure 16: Performance vs $\epsilon$ for LLADA at temperature 0.7

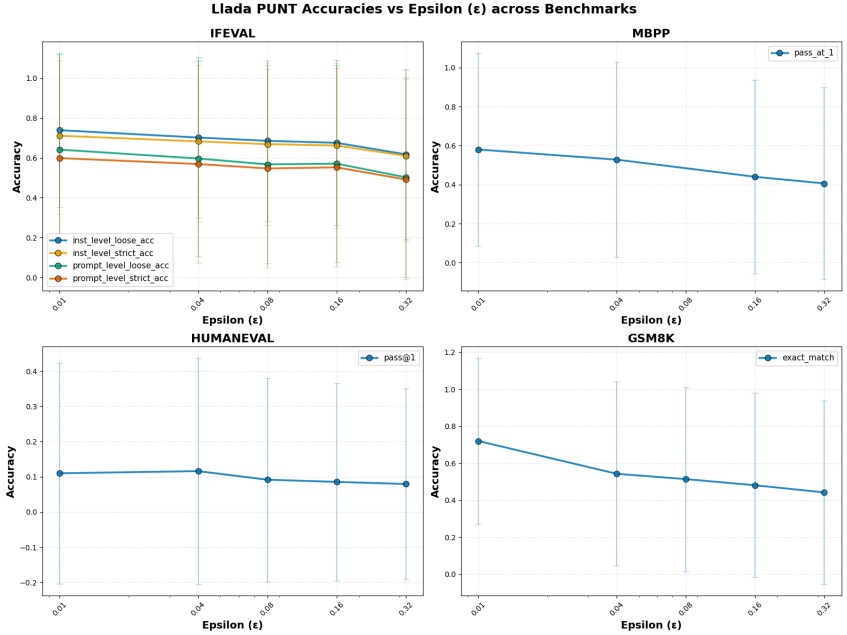

Figure 17: Performance vs $\epsilon$ for DREAM at temperature 0.1

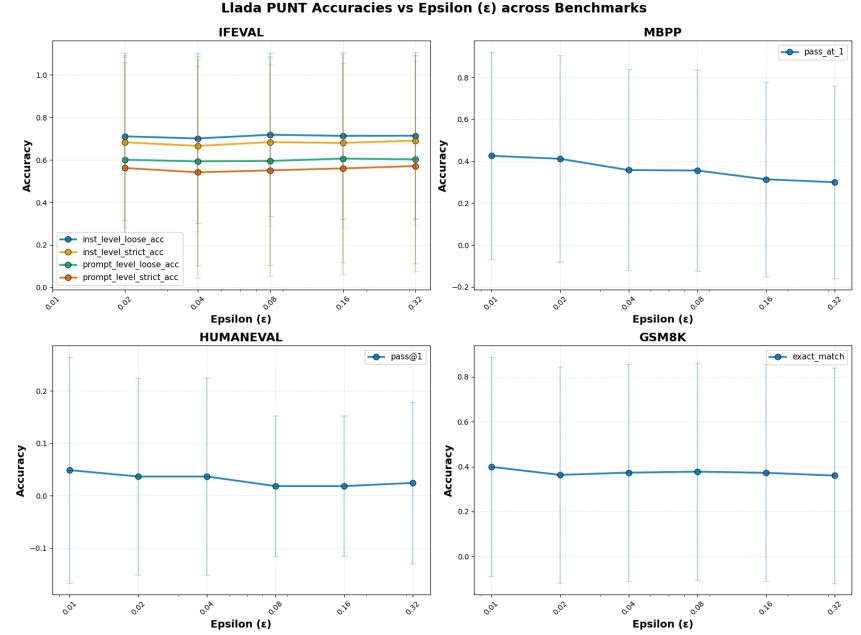

Figure 18: Performance vs $\epsilon$ for DREAM at temperature 0.7

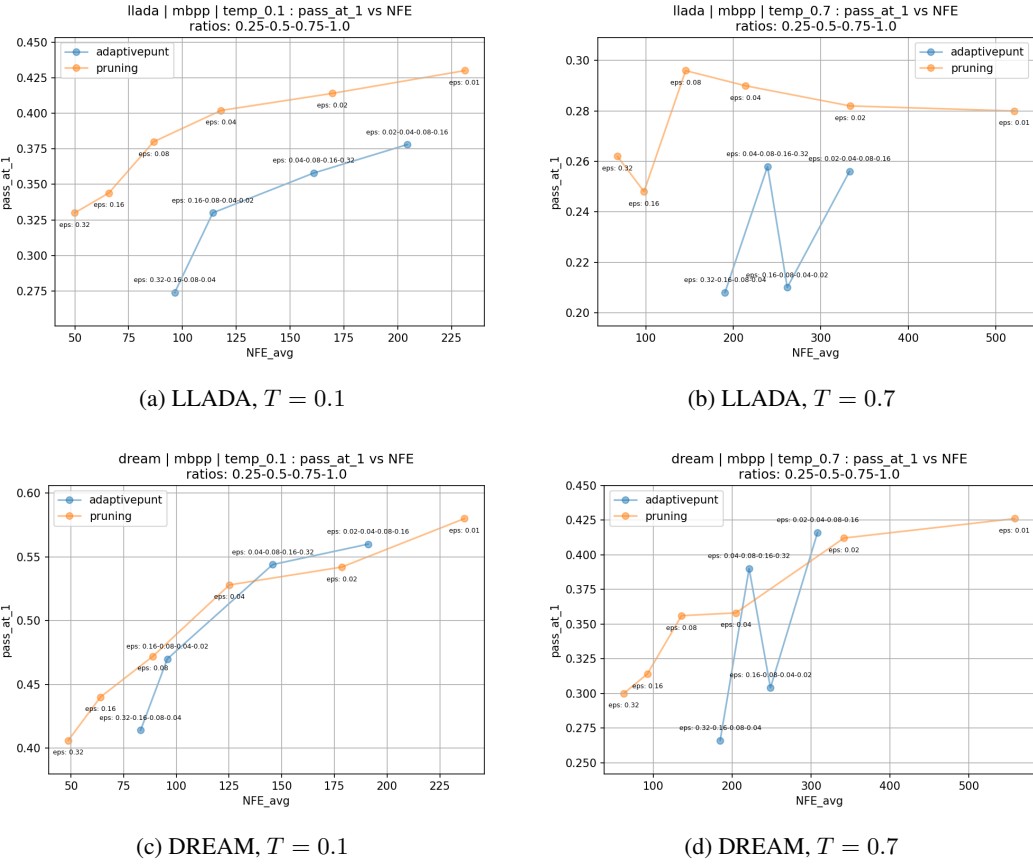

(a) LLADA, $T = 0.1$

(b) LLADA, $T = 0.7$

(c) DREAM, $T = 0.1$

(d) DREAM, $T = 0.7$

Figure 19: MBPP Pass@1 vs NFE for varying epsilon schedules. PUNT with a varying epsilon schedule (blue) vs pruning baseline (orange).

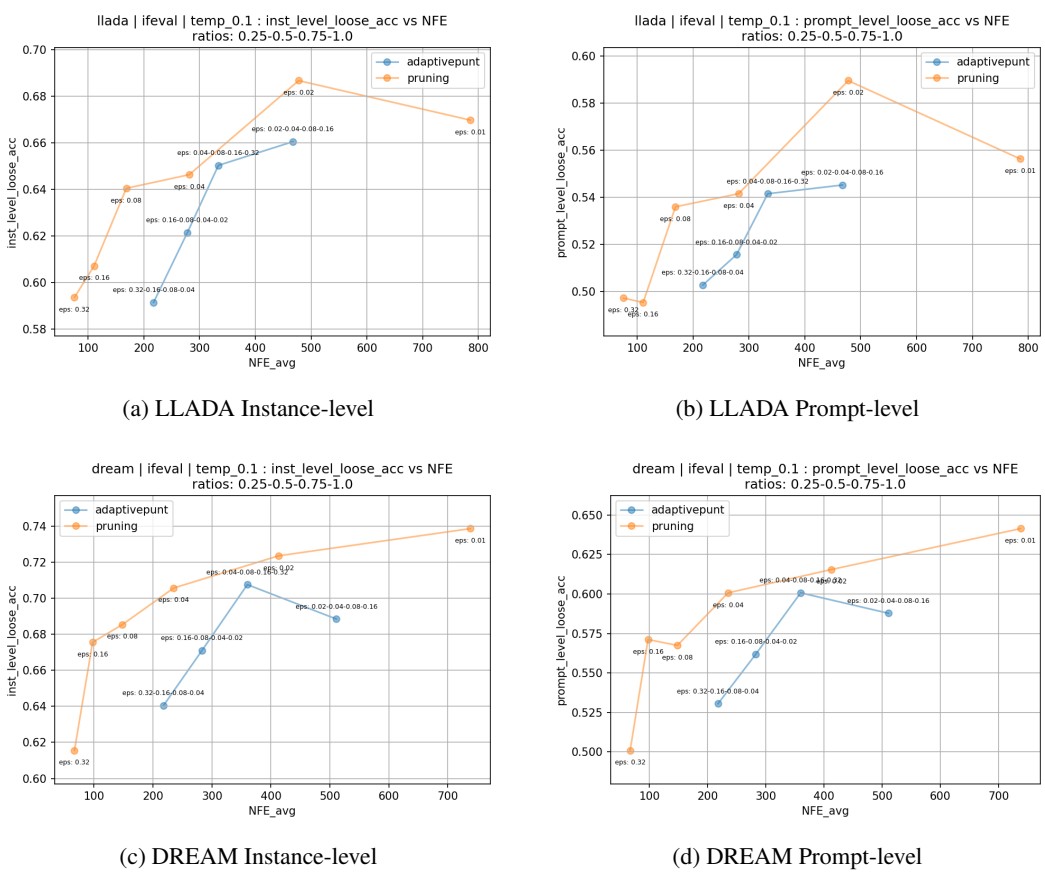

(a) LLADA Instance-level

(b) LLADA Prompt-level

(c) DREAM Instance-level

(d) DREAM Prompt-level

Figure 20: IFEval Loose Accuracy vs NFE at temperature 0.1 for varying epsilon schedules.

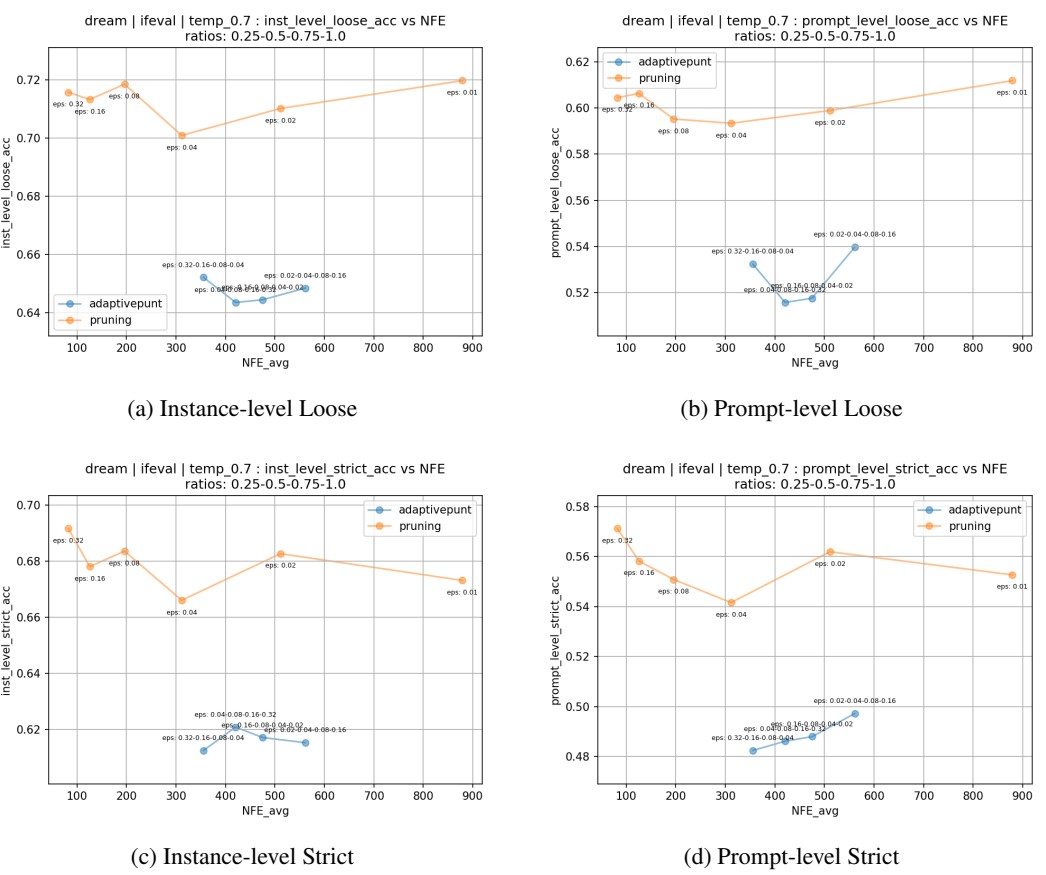

(a) Instance-level Loose

(b) Prompt-level Loose

(c) Instance-level Strict

(d) Prompt-level Strict

Figure 21: DREAM IFEval Accuracy vs NFE at temperature 0.7 for varying epsilon schedules.

