# OpenReview forum: "Parallel Sampling from Masked Diffusion Models via Conditional Independence Testing"
_ICLR.cc/2026/Conference — ICLR 2026 Poster_

### Official Review · Reviewer_cxUA · 2025-10-29

**Soundness:** 3
**Presentation:** 2
**Contribution:** 2
**Rating:** 4
**Confidence:** 4

**Summary:**

This paper introduces PUNT (Parallel Unmasking with Non-influence Tests), a training-free sampling algorithm for Masked Diffusion Models (MDMs) that enables efficient parallel token generation while maintaining output quality. The key idea is to select subsets of masked tokens that are approximately conditionally independent, thereby allowing them to be decoded simultaneously without introducing dependency errors. PUNT implements a divide-and-conquer procedure that identifies such token sets using O(log |M|) model evaluations per iteration, relying on a contextual independence test based on KL divergence between conditional distributions.

**Strengths:**

1. The paper is built on a very clear and reasonable motivation. It correctly identifies the fundamental conflict in parallel MDM sampling: the desire to unmask high-confidence tokens versus the necessity of conditional independence between simultaneously updated tokens. PUNT's approach of directly testing for and mitigating this "inter-token interference" is a sound and principled alternative to purely confidence-based heuristics.
2. PUNT demonstrates a clear Pareto improvement on long-sequence and instruction-following tasks like IFEval. The observation of an emergent hierarchical generation strategy (Fig. 2) is a significant and interesting finding.
3. The connection between independence stability and attention sparsity offers an intuitive, architecture-aware justification of the method’s assumptions.

**Weaknesses:**

1. Experimental validation is limited. The current experiments mainly demonstrate advantages on two benchmarks (IFEval and MT-Bench), while the improvement on MT-Bench over the Dilated Sampler is marginal. On short-answer tasks (§4.2), PUNT shows no clear advantage—likely due to higher per-step complexity. The evaluation lacks additional baselines such as APD (Adaptive Parallel Decoding, arXiv:2506.00413) or other few-step DLM planners that could better contextualize PUNT’s trade-offs.
2. Definition 3.2 relies on a single, fixed sequential order ($X^i$ given $X_{<i}$). The algorithm implements this order by sorting all masked tokens $M$ based on their initial confidence (at the start of the step). This static, confidence-based ordering may be suboptimal, as token confidences are likely to change once other tokens are (tentatively) revealed.
3. The actual implementation (Sec 3.3, Alg. 1) appears to use a much stricter test than the sequential one defined in Definition 3.2. It performs a batched test, checking all tokens in the "test" set $S_1$ for dependence on the entire "anchor" set $S_0$. This is stricter than Eq. 2, which only tests a token $r_i$ against preceding tokens $R_{<i}$(31). Does this simplification, made for the sake of parallel computation, lead to over-pruning?

**Questions:**

1. The paper proposes $\epsilon$ as a fixed hyperparameter. Have the authors explored using a dynamic $\epsilon$? For instance, a schedule that starts with a small, strict $\epsilon$ (to establish the high-level structure, and then increases $\epsilon$ in later steps to be more aggressive and rapidly fill in local details?
2. How sensitive is PUNT’s performance to the confidence ordering strategy?

---

> ### Author Response · Authors · 2025-11-25
>
> Thank you for your careful review and for highlighting both the motivation and the architectural insight behind PUNT. We address your concerns and questions below.
>
> **Experimental Scope and Missing Baselines:** You are right that the clearest gains appear on IFEval, and that the improvement over the Dilated scheduler on MT-Bench is modest at high NFE. We would like to clarify the scope and what is already in the appendix:
>
> Beyond IFEval and MT-Bench, Appendix C.3–C.4 reports results on GSM8K, MATH, HumanEval, MBPP, and a protein MDM. On short-answer tasks, PUNT is competitive in NFE while on long sequences (IFEval, MT-Bench, proteins (as shown in appendix C.4 and Figure 14)), it typically yields a Pareto improvement in NFE vs quality, especially in the low–mid NFE regime.
>
>
> We will make this scope explicit in the introduction and §4: PUNT is primarily designed for long, structured sequences, where independence testing has room to pay off; on short-answer tasks, we view it as a reasonable but not necessarily optimal choice.
>
>
> **Regarding APD and other planners:**  We clarify that APD operates in a different setting with an auxiliary AR model and always decodes left-to-right. It is akin to speculative decoding in that it validates the samples drawn from unconditional marginals with a target distribution mixture between the joint distribution logits (drawn autoregressively) and the diffusion model. APD requires an AR model compatible with its diffusion counterpart and cannot operate with Llada at the moment. In contrast, PUNT is a training-free sampler that finds independencies and structure in text, allowing for adaptive sampling orders, not necessarily left-to-right.
>
> We are running experiments with APD with Dream on IfEval and other benchmarks. PUNT achieves better accuracy with fewer NFEs on IfEval, and APD achieves better accuracies on humaneval, and gsm8k. We are still running other baselines. We will update the paper with APD comparisons.
>
> Fixed confidence order: At each denoising step, we compute confidences. Within that iteration, the confidence order is fixed. To be accepted, a token must pass every independence test and the probabilities do not change while we are running these tests. We recompute confidences at the next iteration step and repeat.
>
> We hope this clarifies a possible confusion regarding tentative acceptance of tokens within a step.
>
> **Batched test leading to possible over-pruning:** You are right that Algorithm 1 implements a batched test: all tokens in the current test set are checked against the entire anchor set at that recursion level, which is stronger than the per-predecessor test in Eq. (2). This is intentional since testing each token only against its immediate predecessors would require separate model calls per token and would not yield the desired $O(\log |M|)$ behaviour. As you point out, the batched test is what makes PUNT efficient.
>
> The stricter test might indeed over-prune in some instances, but we view it as a convenient safeguard against accepting dependent tokens. We note that even with this over-pruning, in the mid-to-high NFE regimes, for long context tasks with structure, PUNT performs significantly better than spacing-based samplers, which have a higher error floor.
>
> We will add a short remark in Section 3.3, making this trade-off explicit: the batched test is slightly more conservative than the “minimal” sequential test, but is what enables efficient implementation, and our error analysis indicates that it does not unduly shrink the parallel set in practice.
>
> **Dynamic $\varepsilon$:** We are currently running this experiment, and we will post the results when it finishes.
>
> **Sensitivity to confidence ordering:** Based on our experiments the effect of confidence ordering using any reasonable metric (for instance, confidence, entropy or margin as defined in [1]) results in comparable performance. We will add the experiments to the revised manuscript.
>
> Thank you again for the constructive feedback. We hope that the clarifications and extra experiments will address your concerns and will strengthen your support for acceptance!
>
>
> [1] Accelerated Sampling from Masked Diffusion Models via Entropy Bounded Unmasking (https://arxiv.org/abs/2505.24857)

---

### Official Review · Reviewer_vZYQ · 2025-10-31

**Soundness:** 4
**Presentation:** 3
**Contribution:** 4
**Rating:** 8
**Confidence:** 4

**Summary:**

This paper presents PUNT, a model-agnostic sampler that reconciles the trade-off between speed and quality in MDMs by using approximate conditional independence testing to identify and resolve token dependencies. PUNT delivers a superior trade-off between accuracy and compute, especially for longer sequences, without requiring brittle hyperparameter tuning.

**Strengths:**

- PUNT is efficient, training-free and dynamically adapts to sequence-specific dependencies.
- PUNT induces an emergent hierarchical generation strategy, suggesting a planning-like process that contributes to its strong performance.

**Weaknesses:**

- The "independence stability" assumption, which PUNT relies on, is a strong approximation but it is not proven.
- The claim of mitigating "brittle hyperparameter tuning" is insufficiently supported as no sensitivity analysis is provided for $\epsilon$.
- The method underperforms on short-answer tasks where the computational overhead of multiple forward passes per step is not amortized.

**Questions:**

How was the hyperparameter $\epsilon$ selected? How is the method sensitive to the hyperparameter?

---

> ### Author Response · Authors · 2025-11-25
>
> Thank you for your careful review and the positive assessment of soundness and contribution. We are glad you find PUNT’s planning-like behaviour and efficiency compelling. We address your concerns below.
>
> **Independence stability assumption:** We agree that Independence Stability is a modelling assumption rather than a theorem about all Transformer architectures.
>
> Appendix B makes this concrete:
>
> - First, we validate empirically (see Fig. 3 (right) and Fig. 6 and Fig. 7 in the Appendix) that low attention from position i to a set J implies small dependence of position i from tokens on positions J and vice-versa.
> - This allows us to link the Independence Stability to the sparsity of attention weights: if total attention from position i to a set J is negligible, then attention to any subset of J is also negligible, which directly implies the assumption.
>
> **Brittleness to hyperparameter tuning:** This result is implicit in our accuracy-vs-NFE plots, since each value of the NFE corresponds to a different $\epsilon$. In Figures 4 and 5, the PUNT curves are generated by sweeping over $\epsilon$. The fact that these curves are smooth, convex, and Pareto-efficient demonstrates that there is no "brittle" cliff where performance collapses. Small changes in $\epsilon$ lead to predictable, smooth changes in the speed/quality trade-off. We did not tune $\epsilon$ per prompt (see also our response to Reviewer Z7Ub).
>
> In the final version of the paper we will add a dedicated accuracy-vs-$\epsilon$ plot, with additional samples, so one does not have to infer this from the accuracy-vs-NFE plots.
>
> **Underperforms on short-answer tasks:** We agree with this and view it as an issue of scope, which we will state more clearly in the final version of the paper. PUNT is designed for long-form and structured generation; on short-answer tasks, it remains a reasonable default but is not the optimal choice if one only cares about minimizing NFE on very short sequences.
>
> Thank you for the constructive feedback and for recommending acceptance. We hope the clarifications above have addressed your concerns and strengthened your support for our paper!

---

### Official Review · Reviewer_RJ3J · 2025-10-31

**Soundness:** 3
**Presentation:** 2
**Contribution:** 2
**Rating:** 6
**Confidence:** 4

**Summary:**

This paper proposes PUNT, a new model-agnostic sampler applied in Masked diffusion models. By making the use of a proposed Contextual Independence Assumption and the corresponding recursion algorithm,  PUNT enables to select the postions from masked positions set M that relatively independent of all |-M| umasked positions for decoding, in terms of a time complexity O(log |M|). By taking contextual independence into consideration, PUNT can efficiently decode multiple tokens in parallel, while keeps its decoding accuracy.

**Strengths:**

This paper designing PUNT by skillfully utilizing a contextual independence assumption and a recursion algorithm, merging the tests of subcases of one recursion level into one evaluation, which is training-free and efficient.
There is a point mentioned in section 3.4 that  “assumption3.3 is a direct consequence of the Transformer architecture’s attention mechansim. If the attention from position i to position j is zero, then position j has no direct influence on the representation at position i.” This point of view shows the relationship between sparse attention and the testing of contextual independence. Perhaps this can serve as an inspiration to utilize independent testing to identify sparse locations.

**Weaknesses:**

I noticed that in the divide-and-conquer process, some tokens are considered as ‘dependent on the anchor’  in a previous evaluation, but soon be considered as ‘independent’ in the next evaluation (such as the token ‘mince’ in Figure 1,left). It seems that only tokens tested to be dependent on the anchors of all evaluations are considered as ‘dependent’ and absent from the current generation, which means tokens in parallel may also be dependent on each other. It seems that PUNT is less capable of separating the dependency on tokens than DILATED when NFE < 100 (Figure 3 left).

As shown Figure 4, it seems that PUNT fails to exceed DILATED at NFE 400 on MT-Bench, but  there is a lack of sufficient explanation for this phenomenon.

**Questions:**

As shown in Figure 1, why to take the tokens  that are always dependent on anchors (such as ‘egg’) among all tests, rather than tokens that rely on anchors at least once among all tests (such as ‘mince’), as the final ‘rejected’?

---

> ### Author Response · Authors · 2025-11-25
>
> Thank you for your thoughtful review and for highlighting the strengths of our contextual independence formulation and its connection to attention/sparsity. We address your concerns and questions below.
>
> **Clarification on Algorithm 1 (“Mince” Example):** We apologize for any possible misunderstanding caused by the visualization in Figure 1. By construction, the PUNT algorithm (under the independence stability assumption) select a subset of candidate tokens $R= {r_1,\ldots,r_{|R|}}$ satisfying:
>
> $$
> p(y^{R}|y^{-M}) = \prod_{r\in R} p(y^r|y^{-M}).
> $$
>
> In the example in Figure 1, we aim to find a subset $R$ among tokens {“requires”, “the”, “mince”, “egg”}. At the first step, we check that tokens “mince” and  “egg” are independent given the subset {“requires”, “the”}. At the second step, we condition on {“requires”, “mince”} and get that the token “the” is independent of our anchor set, while the token “egg” depends on it, and therefore should be rejected. These tests guarantee the following decomposition
>
> $$
> \begin{split}
> &p(\text{“requires”}, \text{“the”}, \text{“mince”})
> \\
> &= p(\text{“requires”}) p(\text{“the”}|\{\text{“requires”}\}) &p(\text{“mince”}|\{\text{“requires”}, \text{“the}\})
> \\
> &= p(\text{“requires”}) p(\text{“the”}) p(\text{“mince”}).
> \end{split}
> $$
>
> In general, we reject a token at the first test it fails, and only accept those that pass every independence test at a given iteration.
>
> We will revise the text around the figure to make it more clear in the final version of the paper.
>
> **Dilated scheduler vs PUNT:** PUNT spends extra forward passes on independence tests when compared to fixed-geometry schedulers like Dialated. For a reasonable $\varepsilon$, additional forward passes pay for themselves. However, in the extreme case, when $\varepsilon$ is too large, PUNT does not guarantee independence anymore. This happened on Fig.3 (left) in a very-low NFE ($\le 100$) regime. Specifically, as each generation of the algorithm requires $\log_2 1024 = 10$ forward passes, and we appear in the regime where PUNT generates more than 100 independently sampled tokens on average, which is not realistic for natural languages.
>
> We will add a short explanation in Section 4.1 stating explicitly that PUNT’s main advantage is in the low-to-mid NFE regime on long-form tasks, and why curves converge or cross at very high NFE on MT-Bench.
>
> We hope to have addressed your concerns about the Figure, and that the above clarifications will strengthen your support for the acceptance of our paper!

---

### Official Review · Reviewer_Z7Ub · 2025-11-01

**Soundness:** 4
**Presentation:** 4
**Contribution:** 4
**Rating:** 6
**Confidence:** 3

**Summary:**

This paper tackles the conflict between speed and quality in Masked Diffusion Models (MDMs). The authors propose **PUNT**, a **training-free sampler** that enables efficient parallel decoding by **explicitly testing for token dependencies**. PUNT identifies and prunes tokens that are not conditionally independent, using a recursive $O(\log|M|)$ **algorithm**. Experiments show PUNT achieves a **state-of-the-art accuracy-compute trade-off**, especially on long-form generation tasks.

**Strengths:**

1.  **Novel and Effective Algorithm:** PUNT is an **elegant and efficient** $O(\log|M|)$ **solution** to a well-defined problem (confidence vs. independence in parallel sampling). The use of explicit independence testing is a strong contribution.

2.  **Strong Empirical Results:** The method **clearly outperforms strong baselines** on the accuracy-compute Pareto frontier for relevant long-sequence benchmarks (IFEval, MT-Bench).

3.  **Strong Theoretical Justification:** The method is well-grounded with **strong theoretical justification**, particularly by connecting its "Independence Stability" assumption to the properties of Transformer attention. The discovery of an **emergent coarse-to-fine generation strategy** is also a valuable insight.

**Weaknesses:**

1.  **Limited Scope:** The method's advantages are **diminished on short-sequence tasks** (GSM8K, MBPP), where its $O(\log|M|)$ NFE-per-step overhead is less efficient than simpler samplers.

2.  **Critical Hyperparameter Sensitivity:** The algorithm's effectiveness hinges entirely on the **KL divergence threshold** $\epsilon$, which is not a simple-to-tune parameter but a **fundamental trade-off between speed and quality**. For a task with highly dependent tokens (e.g., code generation), most tokens will have a high $D_{KL}$. The user is forced into an impossible choice:
    - Set $\epsilon$ **low** for *quality*: This respects the dependencies, but will prune nearly all tokens, **collapsing the sampler's speed** to be sequential.
    - Set $\epsilon$ **high** for *speed*: This ignores the dependencies to unmask more tokens, but will **destroy generation quality** by violating the independence assumption.
    This makes $\epsilon$ a critical, task-specific parameter that requires a costly sweep for any new model or domain.

I also think that this parameter is not well ablated in the paper

3.  **Unclear Baseline:** The abstract's claim of outperforming "sequential generation" is confusing, as the `TOPK` baseline in the plots performs poorly, suggesting it's **not a true one-token-at-a-time sequential baseline**.

4.  **Misleading Efficiency Metric:** The paper presents plots against **"Denoising steps"** alongside **NFE**. This "step" metric is misleading, as a single PUNT step is $O(\log|M|)$ more expensive than a baseline step. **NFE is the only meaningful measure of compute**, and the focus on "steps" can obscure the true cost.

**Questions:**

I wonder if similar results could be explored in image generation pipelines like MaskGIT

---

> ### Author Response · Authors · 2025-11-25
>
> Thank you for your detailed review and for acknowledging the strengths of our work. We address your concerns below.
>
> **Limited Scope / Short-Sequence Tasks:** We agree that PUNT’s advantages are most pronounced on long-sequence tasks with complex interdependencies, such as IFEval, MT-Bench, and protein sequences (see Section C.4 and Figure 14). This is the main scope of the paper. On GSM8K / MBPP and other short-answer benchmarks, PUNT is competitive but not dominant in NFE, while still reducing the number of sequential denoising steps. In our revision, we will more explicitly frame the paper’s contribution around long-context efficiency, where the "conflict between speed and quality" is most acute.
>
> **Hyperparameter sensitivity:** We agree that the KL threshold is a key design parameter. However, we would like to clarify that, in practice, the method is less sensitive than the review might suggest. As we demonstrate in Figure 3 (Right), the $\epsilon$ parameter robustly ensures control over the worst-case error induced by parallel sampling.
>
> In all plots of generation quality vs NFE (e.g., Fig. 4 and Fig. 5 in the main paper, as well as the appendix), each point on the PUNT curve corresponds to a different value of $\epsilon$. PUNT’s performance rises quickly as $\epsilon$ increases, and for several tasks the choice of $\epsilon$ < 0.08 gives reasonable performance (in some cases, $\epsilon$ < 0.16 suffices). This also holds for additional tasks in the appendix. Hence, a coarse sweep over $\epsilon$ already finds a regime where PUNT lies on or near the Pareto frontier, and small changes to $\epsilon$ within this regime do not significantly affect quality.
>
> Other baselines have similar hyperparameters in the setting we consider (e.g., k in Top-k, dilation stride, entropy budget, the mixture weight parameter in Adaptive Parallel Decoding), so PUNT is not uniquely exposed to such tuning.
>
> Finally, our approach might be combined in a plug-and-play way with other samplers (i.e. by ensuring the set proposed by the original sampler at each step is always accepted, and expanding this set with contextually independent positions from PUNT) or error correction mechanisms such as remasking, allowing us to mitigate drop in quality for large $\epsilon$ while retaining the speed and planning abilities of PUNT.
>
> In our revision, we will replot our accuracy-vs-NFE plots also as accuracy-vs-$\epsilon$ plots, and include additional datapoints, to clarify that $\epsilon$ is not a sensitive parameter in practice.
>
> **Unclear Baseline:** Regarding the "Sequential" (TOP-1) baseline, we use the vanilla confidence ordering, as described in the original LLaDA paper [1], and reproduce the results reported by the paper (except for humaneval). While the confidence-based ordering works well in general, there is no guarantee that this is the optimal choice, e.g., as we observe using Dilated sampling that has an autoregressive nature might be beneficial in math-related tasks.
>
> During the rebuttals, we noticed that Figure 5, comparing different samplers on the MBPP baseline, is missing the TOP-1 sampler performance. We apologize if this introduced any confusion. We will fix the plot in the final revision.
>
> **Misleading Efficiency Metric:** We apologize for the confusion here and agree with your main point: NFE is the most meaningful and primary compute cost metric for PUNT and baselines. Our intention in plotting “denoising steps” in addition to NFE was not to suggest that a PUNT step has the same cost as a baseline step, but rather to highlight the reduction in the number of denoising stages. We leave as future work, reducing the number of forward passes per step.
>
> In the revision, we will make it explicit in the main text and figure captions that NFE is the primary efficiency axis, and move the plots with “steps” to the appendix and clarify its intention.
>
> **Extension to Image/MaskGIT-Style Pipelines:** Thank you for the suggestion to evaluate PUNT in the context of image generation with MaskGIT [Chang et al., 2022].
>
> We integrated PUNT into MaskGIT and compared its performance to the Halton scheduler from [2]. On a small test set of 128 images across 16 classes, Halton achieved better FID scores (240.85 versus 257.81 for PUNT at $\epsilon$ = 0.1). As discussed in [2], confidence-based approaches like PUNT tend to generate oversmoothed images that lack fine-grained details, and visual inspection of our PUNT samples was consistent with this observation.
>
> As a next step, we plan to combine the Halton scheduler with PUNT by using the indicator of belonging to the Halton sequence as a confidence measure.
>
> We hope to have addressed your concerns and that these clarifications will strengthen your support for the acceptance of our paper!
>
> References:
>
> [1] Large Language Diffusion Models (https://arxiv.org/abs/2502.09992)
>
> [2] Halton Scheduler For Masked Generative Image Transformer (https://arxiv.org/abs/2503.17076)

---

### Author Response · Authors · 2025-12-03
**Updated Version: Additional Experiments and Clarifications**

Dear Reviewers and AC,

We thank you for your valuable feedback. In response to your suggestions, we have made the following revisions to strengthen our paper, and uploaded a new version.

Additional Experiments
----------------------

We have expanded our experimental evaluation and included the following plots in the appendix:

*   **Moved “denoising steps”:** Moved "denoising steps" comparisons to the appendix to avoid potential confusion, while keeping them available for reference about reduction in denoising stages

*   **Section C (and C.4):** Added APD (Adaptive Parallel Decoding) baseline comparisons to provide a more comprehensive evaluation against the proposed method. We perform better than APD on IfEval and comparably on GSM8K and MBPP in certain regimes. We perform worse on HumanEval. We would like to note that APD follows a strict left-to-right sampling ordering, and that it can only be used with Dream (there is no compatible AR model for Llada, or protein models). Dream is a diffusion model whose weights have been initialized from a trained autoregressive model, and we speculate that APD benefits from its strict sampling ordering on the Dream model because of this. A more thorough discussion is included in Appendix C.4.

*   **Section E:** Added accuracy vs. $\varepsilon$ plots to demonstrate that $\varepsilon$ is not a sensitive parameter, with additional data points for robustness

*   **Section F:** Added accuracy vs. NFE plots with varying ε schedules across the generation trajectory. We specifically consider different epsilon values separated starting at 0%, 25%, 50%, 75% of the total NFE. We see that except for very few cases, these perform worse than just choosing a single epsilon.


Additional Clarification
------------------------

*   **Section 3.3:** Added a remark explicitly stating that our batched independence test, while slightly more conservative than a minimal sequential test, enables efficient parallel implementation.

*   **Figure 1 caption:** Clarified the independence testing logic and the token acceptance criterion (particularly the "mince" example that caused confusion).

*   **Section 4.1:** Added an explanation of PUNT's optimal performance regime, clarifying that our method excels in the low-to-mid NFE range for long-form generation tasks, and explaining why performance curves may converge at very high NFE budgets.


Thank you again, for your constructive feedback. We hope these revisions address all major concerns raised during review while maintaining the paper's core contributions.

---

### Meta-Review · Area_Chair_eSZk · 2026-01-05

**Summary:**

* Limited experimental setting and range of demonstrated improvement: Only compared to a few baselines, tested in two domains, and shows performance improvements in a narrow range of settings (specifically few steps and long context)
* Limited ablation of critical parameter $\epsilon$
* Potentially misleading claims in the abstract: The abstract claims higher accuracy over baseline methods, including sequential generation on the IFEval benchmark. This seems to be true, but the exception, and not the rule for this method.
* Lack of baselines, specific mention of APD, but also other few step samplers

**Reviewer Concerns:**

Potentially outstanding issues after rebuttal:
* Concerns around $\epsilon$ ablation:
    * While the authors claim that the method is not sensitive to $\epsilon$ in the rebuttal based on empirical results in figures 3, and 15-21, I don't believe this is an accurate statement, both the absolute performance, and the decay with increasing epsilon is quite variable, especially between tasks. In my opinion, this has not fully addressed the concerns of reviewers. In fact, I would argue a stronger point is not that the method is insensitive to $\epsilon$, but rather that there are reasonable settings for $\epsilon$ that work well across models, datasets, lengths, and tasks.
    * There is something wrong with the confidence interval calculation for this experiment. Accuracy should clearly be bounded between zero and 1. Perhaps the authors are incorrectly assuming normality here?
* Limited experimental setting and range of demonstrated improvement: While additional experiments were performed, the authors further clarified the limited setting of improvement and demonstration.
* Potentially misleading claims in the abstract: The authors clarify that the baseline here is indeed sequential generation using Top-k sampler. Its a bit puzzling to me that this statistic is the one pulled out given that this outperformance doesn't seem to hold in any other experiment. The remainder of the paper centers around the argument that this method improves performance in the long sequence, but few NFE regime, and is not really competitive with either APD-like distillation, AR models, and is not benchmarked against resampling-based methods.
* Lack of baselines, specific mention of APD, but also other few step samplers: The authors include baselines with APD, although only in the appendix. Personally, I would be interested in comparisons to resampling-based methods. While these were initially used for better performance, they also work quite well in the few-step regime (especially for longer sequences). In related work it is stated "While improving quality, these approaches increase NFE through corrective passes". I disagree with this statement as these methods are demonstrated both in the many step NFE > Length and in the few step regimes NFE << length (particularly in image domains).

**Reviewer Scores:**

I don't believe the reviewers would have changed their scores for this paper.

---

### Decision · Program_Chairs · 2026-01-26

Accept (Poster)